

# Weakening of Antarctic Stratospheric Planetary Wave Activities in Early Austral Spring Since the Early 2000s: A Response to Sea Surface Temperature Trends

YIHANG HU, WENSHOU TIAN, JIANKAI ZHANG, TAO WANG, MIAN XU

*Key Laboratory for Semi-Arid Climate Change of the Ministry of Education, College of Atmospheric*

*Sciences, Lanzhou University, China*

*Correspondence to: wstian@lzu.edu.cn



**Abstract**
Using multiple reanalysis datasets and modeling simulations, the trends of
Antarctic stratospheric planetary wave activities in early austral spring since the early
2000s are investigated in this study. We find that the stratospheric planetary wave
activities in September have weakened significantly since 2000, which is related to the
weakening of the tropospheric wave sources in the extratropical southern hemisphere.
Further analysis indicates that the trend of September sea surface temperature (SST)
over 20°N-70°S is statistically linked to the weakening of stratospheric planetary wave
activities. Numerical simulations support the result that the SST trend in the
extratropical southern hemisphere (20°S-70°S) and the tropics (20°N-20°S) induce the
weakening of wave-1 component of tropospheric geopotential height in the
extratropical southern hemisphere, which subsequently leads to the decrease in
stratospheric wave flux. The responses of stratospheric wave activities in the southern
hemisphere to stratospheric ozone recovery is not significant in simulations. In addition,
both reanalysis data and numerical simulations indicate that the Brewer-Dobson
circulation (BDC) related to wave activities in the stratosphere has also been weakening
in early austral spring since 2000 due to the trend of September SST in the tropics and
extratropical southern hemisphere.
***Key words***: *Antarctic; Stratospheric planetary wave activities; Tropospheric wave*
*sources; Sea surface temperature*



## 1. Introduction

The stratospheric planetary wave activities have important influences on
stratospheric temperature (e.g., Hu & Fu, 2009; Lin et al., 2009; Li & Tian, 2017; Li et
al., 2018), polar vortex (e.g., Kim et al., 2014; Zhang et al., 2016; Hu et al., 2018) and
distribution of chemical substances (e.g., Gabriel et al., 2011; Ialongo et al., 2012;
Kravchenko et al., 2011; Zhang et al., 2019a). Meanwhile, the stratospheric circulation
modulated by planetary waves can exert impacts on tropospheric weather and climate
(e.g., Haigh et al., 2005; Zhang et al., 2019b) through downward control processes
(Haynes et al., 1991), which is useful for extended forecast by using preceding signals
in the stratosphere (e.g., Baldwin et al., 2001; Wang et al., 2020).
The planetary perturbations generated by large-scale topography, convection and
continent-ocean heating contrast can propagate from the troposphere to the stratosphere
(Charney & Drazin, 1961) and form stratospheric planetary waves. As the land-sea
thermal contrast in the northern hemisphere is larger than that in the southern
hemisphere and produces stronger zonal forcing for the genesis of stratospheric waves,
the majority of attention has been given to wave activities and their impacts on weather
and climate in the northern hemisphere (e.g., Kim et al., 2014; Zhang et al., 2016; Hu
et al., 2018). However, planetary wave activities in the southern hemisphere also play
an important role in heating the stratosphere dynamically (e.g., Hu & Fu, 2009; Lin et
al., 2009), which suppresses Polar Stratospheric Clouds (PSCs) formation and ozone
depletion (e.g., Shen et al., 2020a; Tian et al., 2018). The Antarctic sudden stratospheric
warming (SSW) that occurred in 2002 (e.g., Baldwin et al., 2003; Nishii & Nakamura,





2004; Newman & Nash, 2005) and 2019 (e.g., Yamazaki el al., 2020; Shen et al., 2020a;
Shen et al., 2020b) were associated with significant upward propagation of wave flux.
Such episodes are extraordinarily rare in the history, and the one in 2019 contributed to
the formation of the smallest Antarctic ozone hole on record (WMO, 2019). In addition,
some studies reported that wildfires in Australia at the end of 2019 are related to
negative phase of the Southern Annular Mode (SAM), which was induced by the
extended influence of the SSW event that occurred in September (Lim et al., 2019; Shen
et al., 2020b). In a word, the Antarctic planetary wave activities are important for the
stratosphere-troposphere interactions and climate system in the southern hemisphere.

Long-term observations in the Antarctic stratosphere show a significant ozone

decline from the early 1980s to the early 2000s due to anthropogenic emission of
chlorofluorocarbons (CFCs) (WMO, 2011) and a recovery signal since 2000s because
of phasing out CFCs in response to Montreal Protocal (e.g., Angell and Free, 2009;
Krzyścin, 2012; Zhang et al., 2014; Banerjee et al., 2020). The Antarctic stratospheric
ozone depletion and recovery have important impacts on climate in the southern
hemisphere. The ozone depletion cools the Antarctic stratosphere through reducing
absorption of radiation and leads to the strengthening of Antarctic polar vortex during
austral spring (e.g., Randel & Wu, 1999; Solomon et al., 1999; Thompson et al., 2011).
The anomalous circulation in the Antarctic stratosphere during austral spring exerts
impacts on tropospheric circulations (e.g., intensification of SAM index, poleward shift
of tropospheric jet position and expansion of the Hadley cell edge) in the subsequent
months (e.g., Thompson et al., 2011; Swart & Fyfe, 2012; Son et al., 2018; Banerjee et



al., 2020) and influences the distribution of precipitation and dry zone in the southern
hemisphere (e.g., Thompson et al., 2011; Barnes et al., 2013; Kang et al., 2011).
Following the healing of ozone loss in the Antarctic ozone hole since 2000s (e.g.,
Solomon et al., 2016; Susan et al., 2019), great attention has been paid on possible
impacts of ozone recovery on climate system in the southern hemisphere (e.g., Son et
al., 2008; Barnes et al., 2013; Xia et al., 2020; Banerjee et al., 2020). Son et al. (2008)
implemented the Chemistry-Climate Model Validation (CCMVal) models to predict the
response of the southern hemisphere westerly jet to stratospheric ozone recovery. Based
on the Phase 5 of Coupled Model Intercomparison Projects (CMIP5) models, Barnes et
al. (2013) proposed that the tropospheric jet and dry zone edge no longer shift poleward
during austral summer since the early 2000s due to ozone recovery. Banerjee et al.
(2020) analyzed observations and reanalysis datasets. They found that following the
ozone recovery after 2000, the increase of SAM index and the poleward shifting of
tropospheric jet position as well as the Hadley cell edge all experienced a pause. Their
results suggest that ozone depletion and recovery have made important contributions to
the climate shift that occurred around 2000 in the southern hemisphere.

However, some previous studies have reported zonally asymmetric warming

patterns in Antarctic stratosphere, which are generated by increased planetary wave
activities during austral spring from the early 1980s to the early 2000s (Hu & Fu, 2009;
Lin et al., 2009). Note that the Antarctic stratosphere was experiencing radiative cooling
in the same period due to ozone depletion (e.g., Randel & Wu, 1999; Solomon et al.,
1999; Thompson et al., 2011). The increase in stratospheric planetary wave activities



cannot be explained by ozone decline, because the acceleration of stratospheric
circumpolar wind caused by radiative cooling induces more wave energy to be reflected
back to the troposphere (e.g., Andrews et al., 1987; Holton et al., 2004). Hu & Fu (2009)
attributed the increase in Antarctic stratospheric wave activities to the SST trend from
the 1980s to the 2000s. Their results indicate that in addition to ozone change, other
factors such as SST trend also contribute to climate change in the southern hemisphere.
Moreover, the phase of Interdecadal Pacific Oscillation (IPO) also changed at around
2000 (e.g., Trenberth et al., 2013). SST variation influences Rossby wave propagation
and tropospheric wave sources, and thereby indirectly affects stratospheric wave
activities (e.g., Lin et al., 2012; Hu et al., 2018; Tian et al., 2018). The questions here
are: (1) Has the stratospheric planetary wave activity trend in the southern hemisphere
been shifting since the 2000s? (2) What are the factors responsible for the trend of
Antarctic stratospheric planetary wave activity since the 2000s?

In this study, we reveal the trend of Antarctic planetary wave activity in early

austral spring since the 2000s based on multiple reanalysis datasets. We also conduct
sensitive experiments forced by linear increments of ozone and SST fields since the
2000s to investigate the response of Antarctic planetary activity to above two factors.
The remainder of the paper is organized as follows. Section 2 describes the data,
methods and configurations of model simulations. Section 3 presents the trends of
stratospheric and tropospheric wave activities in early austral spring. Section 4
investigates the connections between the trends of SST and stratospheric wave activities.
Sections 5 discusses the responses of tropospheric wave source and stratospheric wave



activity to SST trend based on simulations. Major conclusions and discussion are
presented in Section 6.

### 2. Datasets, methods and experimental configurations

a.  Datasets

In this study, daily and monthly mean data extracted from the Modern-Era

Retrospective analysis for Research and Applications Version 2 (MERRA-2;
Bosilovich et al., 2015) dataset are used to calculate trends of zonally averaged zonal
wind and temperature, BDC, tropospheric wave sources, and the Elisassen-Palm (E-P)
flux and its divergence in September. To verify the trend of stratospheric E-P flux, we
also refer to the results derived from the European Centre for Medium-range Weather
Forecasting (ECMWF) Interim Reanalysis (ERA-Interim; Dee et al., 2011) dataset, the
Japanese 55-year Reanalysis (JRA-55; Kobayashi et al., 2015) dataset and the National
Centers for Environmental Prediction-Department of Energy Global Reanalysis 2
(NCEP-2; Kanamitsu et al., 2002) dataset.

SST data are extracted from the Extended Reconstructed Sea Surface Temperature

(ERSST) dataset, which is a global monthly mean sea surface temperature dataset
derived from the International Comprehensive Ocean-Atmosphere Dataset (ICOADS).
The ERSST is on global 2°×2° grid and covers the period from January 1854 to the
present. We use the newest version (version 5, i.e., v5) dataset to calculate trends and
correlations, and produce SST forcing field for model simulations. More details about
this version of ERSST can be found in Huang et al. (2017).

In addition, the unfiltered Interdecadal Pacific Oscillation (IPO) index derived

 



from the ERSST v5 dataset is also used in this study. The IPO index is available at https://psl.noaa.gov/data/timeseries/IPOTPI/tpi.timeseries.ersstv5.data and more detailed information about the index can be found in Henley et al. (2015).

b. Diagnosis of wave activities and Brewer-Dobson circulation

Planetary wave activities are measured by E-P flux ($\vec{F} \equiv (0, F^{(\phi)}, F^{(z)})$) and its divergence $D_F$. Their algorithms are expressed by Eqs. (1)-(3) (Andrews et al., 1987):

$$F^{(\phi)} = \rho_0 a \cos\phi (\overline{u}_z \overline{v'\theta'} / \overline{\theta}_z - \overline{v'u'}) \quad (1)$$

$$F^{(z)} = \rho_0 a \cos\phi \{[f - (a\cos\phi)^{-1} (\overline{u}\cos\phi)_\phi] \overline{v'\theta'} / \overline{\theta}_z - \overline{w'u'}\} \quad (2)$$

$$D_F = \frac{\nabla \cdot \vec{F}}{\rho_0 a \cos\phi} = \frac{(a\cos\phi)^{-1} \frac{\partial}{\partial \phi}(F^{(\phi)}\cos\phi) + \frac{\partial F^{(z)}}{\partial z}}{\rho_0 a \cos\phi} \quad (3)$$

where $u, v$ represent zonal and meridional components of horizontal wind, $w$ is vertical velocity, $\theta$ is potential temperature, $a$ is the Earth radius, $f$ is the Coriolis parameter, $z$ is geopotential height, $\phi$ is latitude, $\rho_0$ is the background air density.

The quasi-geostrophic refractive index (RI) is used to diagnose the environment of wave propagation (Chen & Robinson, 1992). Its algorithm is written as Equation (4):

$$RI = \frac{\overline{q}_\varphi}{\overline{u}} - \left(\frac{k}{a\cos\varphi}\right)^2 - \left(\frac{f}{2NH}\right)^2 \quad (4)$$

where the zonal-mean potential vorticity meridional gradient $\overline{q}_\varphi$ is

$$\overline{q}_\varphi = \frac{2\Omega}{a}\cos\varphi - \frac{1}{a^2}\left[\frac{(\overline{u}\cos\varphi)_\varphi}{a\cos\varphi}\right]_\varphi - \frac{f^2}{\rho_0}\left(\rho_0 \frac{\overline{u}_z}{N^2}\right)_z \quad (5)$$

$q$, $k$, $N^2$ and $\Omega$ are the potential vorticity, zonal wave number, buoyancy frequency, and Earth's angular frequency, respectively.

The Brewer-Dobson circulation driven by wave breaking in the stratopause is closely related to stratospheric wave activities. Its meridional and vertical components

$(\bar{v}^*, \bar{w}^*)$ and stream function ($\psi^*(p,\phi)$) are expressed by Eqs. (4)-(6) (Andrews et al.,
1987; Birner & Bönisch, 2011) :

$$\bar{v}^* \equiv \bar{v} - \rho_0^{-1}(\rho_0 \overline{v'\theta'} / \overline{\theta_z})_z \tag{6}$$

$$\bar{w}^* \equiv \bar{w} + (a\cos\phi)^{-1}(\cos\phi \cdot \overline{v'\theta'} / \overline{\theta_z})_\phi \tag{7}$$

$$\psi^*(p,\phi) = \int_0^p \frac{-2\pi a \cdot \cos\phi \cdot \bar{v}^*(p'',\phi)}{g} dp'' \tag{8}$$

where $p$ is the air pressure, $\pi$ is the circular constant, $g$ is the gravitational
acceleration.
In Eqs. (1)-(8), the overbar and prime denote zonal mean and departure from zonal
mean, respectively. The subscripts denote partial derivatives. The Fourier
decomposition is used to obtain components of Eqs. (1)-(3) with different zonal wave
numbers. Meanwhile, the Fourier decomposed components of geopotential height zonal
deviations are also used to determine tropospheric wave sources.
c. Statistical methods
The trend is measured by the slope of linear regression based on the least square
estimation. The correlation is used to analyze statistical links between different
variables. In this paper, all the time series have been linearly detrended before
calculating correlation coefficients ($r$) and their corresponding significances.
The change-point testing (e.g. Banerjee et al., 2020) is used to make sure the
significance of trend or correlation coefficient is not unduly influenced by some
particular beginning or ending years, and thereby confirm that the trend exists
objectively.
We use two-tailed student's t test to calculate the significances of trend, correlation





coefficient or mean difference. The result of significance test is measured by p value or
confidence intervals in this paper. $p \leq 0.1$, $p \leq 0.05$ and $p \leq 0.01$ suggest the trend,
correlation coefficient or mean difference is significant at/above the 90%, 95% and 99%
confidence levels, respectively. The confidence interval of trend is shown in (7):
$$[\hat{b} - t_{1-\alpha/2}(n-2)\hat{\sigma}_b, \hat{b} + t_{1-\alpha/2}(n-2)\hat{\sigma}_b] \tag{7}$$
where $\hat{b}$ is estimated value of slope, $\hat{\sigma}_b$ is standard error of slope and it is written
as: $\hat{\sigma}_b = \hat{b} \cdot \sqrt{\dfrac{\frac{1}{r^2} - 1}{n-2}}$, $t_{1-\alpha/2}(n-2)$ denotes the value of t-distribution with the degree
of freedom equal to $n-2$ and the two-tailed confidence level equal to $1-\alpha$
($\alpha = 0.90$, $0.95$ or $0.99$). The confidence interval of mean difference is expressed
by Eq. (8):
$$[\bar{X} - \bar{Y} - t_{1-\alpha/2}(M+N-2) \cdot S_w \cdot \sqrt{\frac{1}{M} + \frac{1}{N}}, \bar{X} - \bar{Y} + t_{1-\alpha/2}(M+N-2) \cdot S_w \cdot \sqrt{\frac{1}{M} + \frac{1}{N}}] \tag{8}$$
where
$$S_w = \sqrt{\frac{1}{M+N-2}[\sum_{i=1}^{M}(X_i - \bar{X})^2 + \sum_{j=1}^{N}(Y_j - \bar{Y})^2]} \tag{9}$$
Here, $\bar{X}$ and $\bar{Y}$ are the sample averages, $M$ and $N$ are the numbers of sample
sizes with two populations, $t_{1-\alpha/2}(M+N-2)$ denotes the value of t-distribution with
the degree of freedom equal to $M+N-2$ and the two-tailed confidence level equal
to $1-\alpha$.

Previous studies have indicated that SST impact on the stratosphere shows a

spatial dependence (e.g. Xie et al., 2020). To find out a robust relationship between the
trend of SST in a specific region and the trend of stratospheric wave activities, we divide


the global ocean into three regions: SH (the extratropical southern hemisphere, 70°S-
20°S), TROP (the tropics, 20°S-20°N) and NH (the extratropical northern hemisphere,
20°N-70°N). Since the impacts in different regions might be combined, we also
consider three combined regions named as SHtrop (the extratropical southern
hemisphere and the tropics, 70°S-20°N), NHtrop (the extratropical northern hemisphere
and the tropics, 20°S-70°N) and the Globe (70°S-70°N). To find statistical connections
between the trend of SST and that of stratospheric wave activities, we examine the first
three leading patterns (EOF1, EOF2, EOF3) and principal components (PC1, PC2, PC3)
of SST in above six regions obtained from Empirical Orthogonal Function (EOF)
analysis. In all the six regions, there is always one EOF modes that shows great
similarity to the spatial pattern of trend (not shown) as we do not detrend SST time
series when the EOF analysis is carried out. Thus, the significance of the correlation
between the PC time series of that EOF mode and time series of stratospheric E-P flux
can be used as the criterion to determine the statistical connection between the trend of
SST and the trend of stratospheric wave activities.
d.   The model and experiment configurations

The FWSC component in the Community Earth System Model (CESM; version

1.2.0) is used to verify the impact of SST and ozone recovery trends on tropospheric
wave sources and stratospheric wave activities in early austral spring. The FWSC
component is the Whole Atmosphere Community Climate Model version 4 (WACCM4)
with specified chemistry forcing fields (such as ozone, greenhouse gases (GHG),
aerosols and so on), which have fixed values in 2000 by default. The WACCM4





includes active atmosphere, data ocean (run as a prescribed component, simply reading
SST forcing data instead of running ocean model), land and sea ice. Important physics
schemes in the WACCM4 are based on those in the Community Atmospheric Model
version 4 (CAM4; Neale et al., 2013). The WACCM4 uses a finite-volume dynamic
framework and extends from the ground to approximately 145 km ($5.1 \times 10^{-6}$ hPa)
altitude in the vertical with 66 vertical levels. The simulations presented in this paper
are conducted at a horizontal resolution of 1.9°×2.5°. More information about the
WACCM can be found in Marsh et al. (2013).

Control experiments and sensitive experiments are conducted to investigate

responses of Antarctic stratospheric wave activities to SST trend and the ozone recovery
trend in early austral spring. For the experiments of SST trends, monthly mean global
SST during 1980-2000 derived from the ERSST v5 dataset is used as SST forcing field
in the control experiment (sstctrl). For the four sensitive experiments (sstNH, sstSH,
ssttrop, sstSHtrop), linear increments of SST in different regions in September during
2000-2017 are used as the forcing field. Ozone, aerosols and greenhouse gases (GHG)
in the control experiment and the four sensitive experiments all have the fixed values
in 2000. For the experiments of ozone recovery trend, monthly mean three-dimensional
global ozone during 1980-2000 derived from the MERRA-2 dataset is used as the ozone
forcing field in the control experiment (O3ctrl). The sensitive experiment (O3sen) is
forced by linear increments of ozone in September during 2001-2017. The SSTs in
O3ctrl and O3sen both are monthly mean global SST during 1980-2000. The aerosol
and greenhouse gases values in 2000 are used. These experiment configurations are



summarized and listed in Table 1 and Table 2.
First, we run the FWSC component to generate randomly different initial
conditions for 120 years with free run. Then, each experiment includes 100 ensemble
members that run from July to September forced by these initial conditions from the
21st year to the 120th year in July. The forcing fields of SST and ozone are only
superposed from July to September. July and August are taken as spin-up time and
simulations during this period are discarded. The ensemble mean in September derived
from these 100 ensemble members is regarded as the final result of each experiment. A
similar approach is implemented for sensitive experiments, in which the forcing fields
superposed only in certain months. The same approach has been used in previous
studies (e.g., Zhang et al., 2018).
**3. Trend of planetary wave activities in early austral spring**
Figure 1 shows the trends of stratospheric planetary wave activities in the southern
hemisphere September during 1980-2000 and 2000-2017, respectively. Note that the
vertical E-P flux entering into the stratosphere over 50°S-70°S in September has been
increasing during 1980-2000, accompanied by intensified wave flux convergence in the
upper stratosphere (Fig. 1a) that is mainly contributed by the wave-1 component (Fig.
1b). This feature implies that the stratospheric planetary wave activities have
strengthened in early austral spring during 1980-2000. A similar result has been
reported in previous studies (Hu & Fu, 2009; Lin et al., 2009). During 2000-2017,
however, vertical transport of stratospheric E-P flux weakened over the subpolar region
of the southern hemisphere, which was accompanied by intensified wave flux



divergence in the upper stratosphere (Fig. 1d) mainly contributed by the wave-1
component (Fig. 1e) while the wave-2 component also made certain contributions (Fig.
1f). Similar features also appear in August, but not as significant as that in September
(Fig. S1). For this reason, hereafter we focus on the features in September.
The SSW that occurred in 2002 was accompanied with large upward wave fluxes
in the stratosphere, which is extremely rare in history and has been studied in numerous
previous studies (e.g., Baldwin et al., 2003; Nishii & Nakamura, 2004; Newman &
Nash, 2005). Since the period with a negative trend of stratospheric vertical wave flux
is short, it is necessary to further investigate whether such a negative trend is artificially
influenced by the single year of 2002. Therefore, following Banerjee et al. (2020), we
use a change-point method to test the significance of the trend during various periods
based on four reanalysis datasets (ERA-Interim, MERRA-2, JRA-55, NCEP-2).
Figures 2a (including the year 2002) and 2b (excluding the year 2002) display the time
series (Fz) of area-weighted vertical stratospheric wave flux over the southern
hemisphere subpolar region obtained from different reanalysis datasets. Note that the
wave flux time series obtained from the four reanalysis datasets all present a positive
trend from the early 1980s to the early 2000s and a negative trend from the early 2000s
to present, regardless of whether the extreme value in 2002 is removed or not. The
correlation coefficients of the time series between these reanalysis datasets are above
0.9 and statistically significant (Table 3), suggesting that the time series derived from
different datasets are consistent with each other. Figures 2c-f show the trends and
corresponding confidence intervals calculated with four different beginning years (1980,





1981, 1982, 1983), four different ending years (2015, 2016, 2017, 2018), and change-
point years from 1998 to 2013. The trends and confidence intervals in Figures 2g-j are
the same as that in Figures 2c-f, except that the extreme value in 2002 is removed. The
positive trend from the early 1980s to the 21st century remains significant regardless of
different beginning years and ending change-point years (Figs. 2c-j). However, Figures
2c-f and Figures 2g-j indicate that the positive value of the trend is decreasing gradually
when the period is prolonged, which is apparently attributed to the negative trend with
the beginning change-point year of around 2000. Although the negative trend from the
change-point year to ending year becomes less significant when the value in 2002 is
removed, it remains significant in some periods, which are also illustrated on diagrams
of latitude-pressure profiles (Fig. S2). Therefore, the weakening of stratospheric wave
activities in early austral spring since the early 2000s is robust. In this paper, we take
the year 2000 as the beginning year of the weakening trend to simplify descriptions in
the following discussion.

Figure 3 shows the trends of tropospheric wave sources in September since 2000.

There is a significant positive trend of the wave-1 component in 500 hPa geopotential
height over the southern Indian ocean and a significant negative trend over the southern
Pacific, which form an out-of-phase superposition on its climatology (Fig. 3b). The
trend pattern of wave-2 component is also out-of-phase with its climatology, although
it is not significant (Fig. 3c). The above features still maintain when the values in 2002
are removed (Fig. S3b, c), implying that the southern hemispheric tropospheric wave
sources in early austral spring have weakened since 2000, which is also reflected in the



decrease of tropospheric vertical wave flux (Fig. 3d, e; Fig. S3d, e).

## 4. Role of SST trends in the weakening of Antarctic stratospheric wave activities

In this section, we further explore factors that lead to the weakening of
tropospheric wave sources and stratospheric wave activities since the early 2000s in
early austral spring. Numerous studies reported that the variations in sea surface
temperature can affect stratospheric climate (e.g., Li, 2009; Hurwitz et al., 2011; Lin et
al., 2012; Hu et al., 2014; Hu et al., 2018; Tian et al., 2018; Xie et al., 2020). Hu & Fu
(2009) also attributed the strengthened stratospheric wave activities in the southern
hemisphere to SST trend from the early 1980s to the early 2000s. Furthermore, global
SST in September during 2000-2017 also has a significant trend. The significant
warming pattern is mainly found over the southern Indian ocean, the southern Atlantic
ocean, the eastern and western equatorial Pacific, the western equatorial and Northern
Atlantic ocean (Fig. 4b). A significant cooling pattern is located over the southeast
Pacific (Fig. 4b). In a word, the spatial pattern of SST trend during 2000-2017 is
obviously different from that during 1980-2000 (Fig. 4a, b). Thus, it is necessary to
analyze the connection between SST trend and wave activity trend since the early 2000s.
Figure 5 shows the significance of principle component (PC) trends (Figs. 5a-f) of
SST in different regions, and the significance of correlations (Figs. 5g-l) between the
PC time series and Fz during various periods in September. The trend of PC1 time series
in SH region is significant during serval periods (Fig. 5a), while the correlation between
PC1 and Fz is only significant with the particular ending year of 2015 (Fig. 5g). This



feature suggests that the connection between the SST trend in SH region and the trend
of stratospheric wave activity is not robust. The correlation between trend of
stratospheric wave activity and that of SST in TROP or NH region is also weak (Fig.
5e, f). As for the combined regions, note that the PC2 time series in SHtrop region has
a significant trend (Fig. 5d) and the correlation between the PC2 time series in SHtrop
and Fz with the beginning year of around 2000 is also significant (Fig. 5j) regardless of
different ending years. This feature implies that the extratropical southern hemisphere
and tropical SST has a robust connection with stratospheric wave activities in early
austral spring since the early 2000s. The correlations between Fz and all PC time series
in NHtrop (Fig. 5k) and Globe (Fig. 5l) region are not as robust as that between Fz and
PC2 time series in SHtrop region (Fig. 5j), indicating that the connection between SST
trend in extratropical northern hemisphere and the trend of stratospheric wave activity
is weak.

Figure 6 shows the first three EOF modes of September SST in SHtrop region

during 2000-2017. The second mode (Fig. 6b) shows a great similarity to the spatial
pattern of SST trend (Fig. 4b), and the corresponding PC2 time series also has a
significant trend (slope=1.71, p<0.01). The correlation between PC2 and the Fz is
significant (r=-0.56, p=0.016) and the correlation coefficient remains significant (r=-
0.46, p=0.065) at the 90% confidence level when the value in 2002 is removed. This
result suggests that the SST trend in SHtrop region is closely related to the recent
weakening of stratospheric wave activities. The first EOF mode is similar to IPO (Fig.
6a) and its corresponding principal component is highly significantly correlated (r=-





0.98, p<0.01) with the unfiltered IPO index. However, it shows no significant trend (Fig.
6d) and has no significant correlation (Fig. 6g) with stratospheric wave flux, implying
that the linkage between the IPO phase change at around 2000 (e.g. Trenberth et al.,
2013) and the weakening of Antarctic stratospheric wave activities is weak. The
correlation between PC3 and Fz is also not significant (Fig. 6i). Therefore, it is possible
that the combined effect of SST trend (the second EOF mode) in the tropical and
extratropical southern hemisphere leads to the weakening of stratospheric wave
activities in early austral spring since the early 2000s.
**5.  Simulated changes in Antarctic stratospheric wave activities forced**

**by SST trends**

The analysis in Section 4 suggests that the SST trend in SHtrop region may

contribute to the weakening of the southern hemispheric stratospheric wave activities.
Here, numerical experiments sstNH, sstSH, ssttrop and sstSHtrop forced by linear
increments of SST in September during 2000-2017 (Fig. 7; more details can be found
in Section 2) are conducted to verify the results discussed in Section 4.

Figure 8 shows the simulated response of 500 hPa geopotential height to SST

changes in different regions. The climatological distributions of the wave-1 component
(Figs. 8b, e, h, k) and the wave-2 component (Figs. 8c, f, i, l) from the simulations are
consistent with that from reanalysis dataset (Figs. 4b, c), indicating that the model can
well capture spatial distributions of the atmospheric waves. Note that the Fourier
component (wave-1 and wave-2) anomalies simulated with SST changes in SH, TROP
and SHtrop all are significant. They superpose on the corresponding climatological





patterns in an out-of-phase style (Figs. 8e, f, h, i, k, l), indicating that the SST trends in
SH, TROP and SHtrop lead to a weakening of tropospheric wave sources in the
extratropical southern hemisphere. However, the 500 hPa geopotential height anomaly
of the predominate wave-1 component in the extratropical southern hemisphere forced
by the experiment with NH SST change is relatively weak (Fig. 8b). This feature
suggests that the SST trend in extratropical northern hemisphere is incapable of
inducing a robust response of tropospheric wave sources in the extratropical southern
hemisphere.
Figure 9 shows the simulated responses of stratospheric wave activities in the
southern hemisphere to SST changes in different regions. It is found that the
experiments with SST changes in SH, TROP and SHtrop show significantly weakened
stratospheric wave activities (Figs. 9d, g, j), which are mainly attributed to the responses
of the wave-1 component (Figs. 9e, h, k). These results are consistent with the responses
of tropospheric wave sources (Figs. 8d, e, g, h, j, k). However, there are no significant
anomalies of stratospheric wave flux in the subpolar region as exhibited in Figures 9a
and 9b, which is consistent with the response of corresponding tropospheric wave
sources (Fig. 8a, b) and the weak correlation between Fz and PC time series of SST in
NH region (Fig. 5i). It suggests that the response of southern hemisphere stratospheric
wave activities to SST trend in NH region is weak.
Results of all these experiments are summarized and displayed in Figure 10, which
is quantified by the frequency distribution of southern hemisphere stratospheric vertical
wave flux derived from the 100 ensemble members of each experiment. Compared to



the blue fitting curves, the red fitting curves shift to the left as shown in Figs. 10b, 10c
and 10d, suggesting that the SST changes in SH, TROP and SHtrop regions weaken the
upward propagation of stratospheric wave flux. The area-weighted anomalies of
vertical E-P flux in the subpolar region of the southern hemisphere induced by SST
changes in SH, TROP and SHtrop regions are $-0.084 \times 10^5$ kg·s$^{-2}$, $-0.12 \times 10^5$ kg·s$^{-2}$ and
$-0.13 \times 10^5$ kg·s$^{-2}$, respectively. The sum of the anomalies forced by sstSH and ssttrop is
not equal to the anomaly forced by sstSHtrop, which may be resulted from non-linear
interactions between the responses of wave activities to SST trends in SH region and
TROP region. The weakening of stratospheric wave activities forced by SST increment
in the tropical region is more obvious and more significant than that in extratropical
southern hemisphere (Figs. 10b, c, e), implying that the SST trend in the tropical region
contributes more to the weakening of stratospheric wave activities since 2000.
Meanwhile, it is apparent that the weakening of the southern hemisphere stratospheric
wave activities forced by sstSHtrop is the most significant among all the sensitive
experiments (Fig. 10e). The reduction of vertical E-P flux over (50°S-70°S, 200 hPa-
10 hPa) forced by sstSHtrop is approximately 12%. These simulation results indicate
that the weakening of the Antarctic stratospheric wave activities in September since
2000 is induced by the combined effects of SST trends in the tropical and extratropical
southern hemisphere. It also explains why the independent correlation between Fz and
PC obtained for SH or TROP region is not as significant as that between Fz and PC
obtained for SHtrop region (Figs. 5g, h, j). Moreover, the mean linear increment of area-
weighted vertical E-P flux from 200 hPa to 10 hPa over 70°S-50°S in September during





2000-2017 derived from four reanalysis datasets is about $-0.38 \times 10^5$ kg·s$^{-2}$. Therefore,
the contribution of SST trend over 20°N-70°S (the SHtrop region) to the weakening of
stratospheric activities is approximately 34%.
**6. Conclusions and Discussion**
This study analyzes the trend of Antarctic stratospheric planetary wave activities
in early austral spring since the early 2000s based on various reanalysis datasets. Using
the change-point method, we find that the Antarctic stratospheric wave activities in
September have been weakening significantly since 2000, which means the intensified
trend of wave activities noted in previous researches (Hu & Fu, 2009; Lin et al., 2009)
are reversed after 2000 in early austral spring. Further analysis suggests that the
weakening of stratospheric wave activities is related to the weakening of tropospheric
wave sources in extratropical Southern Hemisphere, which is mainly contributed by the
wave-1 component. Moreover, EOF analysis and correlation analysis indicate that the
stratospheric wave activities in early austral spring during 2000-2017 are related to PC2
of SST over 20°N-70°S (i.e., the SHtrop region). The corresponding EOF2 mode also
shows a great similarity to the spatial pattern of SST trend, suggesting that the
weakening of stratospheric wave activities is connected to the trend of SST in SHtrop
region. Meanwhile, the linkage between the SST trend in NH region and the weakening
of stratospheric wave activities is weak. Finally, the model simulations support the
conclusion that the SST changes in SHtrop region lead to the weakening of tropospheric
wave sources and stratospheric wave activities. The contribution of SST trend in
tropical region to the weakening of stratospheric wave activities is larger than that in



the extratropical southern hemisphere. However, the response of tropospheric wave
sources and stratospheric wave activities to SST trend in NH region is not significant.
The contribution of SST trend over SHtrop region to the weakening of stratospheric
wave activities is about 34%.

The question that remains answered is whether the ozone recovery trend also

contributes to the weakening of stratospheric wave activities in September since the
early 2000s. As described in Section 2, a control experiment (O3ctrl) forced by
climatological ozone and a sensitive experiment forced by the linear increment of
global ozone in September during 2001-2017 are conducted to address the above
question. The pattern of ozone forcing field is similar to its trend pattern (Figs. S4c, d;
Fig. S5). We choose the period of 2001-2017 because we notice that the ozone recovery
trend derived from MERRA-2 in September with the beginning year of 2000 is not
significant (Fig. S4a, b). Meanwhile, as the SSW in 2002 induces poleward transport
of large amounts of ozone, the data in 2002 are removed when linear increments are
calculated. Other details about these two experiments have been given in Section 2 and
Table 2. The simulated results indicate that there is no significant response of wave flux
(Fig. 11a, d) as well as its Fourier decomposed components (Fig. 11b, c) over southern
hemisphere subpolar region in the stratosphere, suggesting that the prescribed ozone
recovery is incapable of inducing the weakening of stratospheric wave activities.

Many researchers claimed that the climate transition around 2000 in the southern

hemisphere is related to ozone depletion and recovery (e.g., Barnes et al., 2013;
Banerjee et al., 2020). Note that there is no contradiction between our results and these



previous studies. Firstly, the southern hemisphere tropospheric circulation (i.e., the
SAM index, the tropospheric jet position and the Hadley cell edge) transition related to
ozone depletion and recovery reported in these previous studies basically occurred in
austral summer (e.g., Son et al., 2008; Thompson et al., 2011; Barnes et al., 2013;
Banerjee et al., 2020). These tropospheric circulation transitions are induced by
downward coupling of circulation anomalies in the stratosphere (e.g., Thompson et al.,
2011) during October and November, when solar radiation covers the entire Antarctic
and causes radiative heating effects. However, we focus on September in the present
study. The Antarctic stratospheric circulation response to ozone variation in September
is not as strong as that in October or November (e.g., Thompson et al., 2011, Fig. 1b, d)
because solar radiation can only reach part of the Antarctic stratosphere during a
majority period of September. This fact implies that the response of wave propagation
environment in the Antarctic stratosphere to ozone trend is also not significant (Fig. S6).
Secondly, the FWSC component used in this study is an atmospheric module with
prescribed SST and gases. Therefore, the model results only indicate that the weakening
of stratospheric wave activities can be attributed to SST trends, while the impact of
ozone depletion and recovery trend in the tropics and mid-latitudes on the shift of SST
trend pattern cannot be determined based on the model simulations. This is an issue
beyond the scope of this study and further investigation is necessary using a fully
coupled earth system model.

In addition, the reanalysis datasets show that the Brewer-Dobson circulation

related to wave activities in the stratosphere weakened significantly in early austral





spring during 2000-2017 (Fig. 12b), which is contrary to the intensified trend during
1980-2000 (Fig. 12a). The transition of BDC around 2000 is believed to be associated
with ozone depletion and recovery (e.g., Polvani et al., 2017; Polvani et al., 2018).
However, our modeling results suggest that the SST trend is responsible for the
weakening of BDC in September since 2000 (Fig. 12d, e, f), The response of BDC to
ozone recovery is not significant (Fig. 12c), especially for the branch near the Antarctic.
These results indicate that the SST trend should be taken into consideration when
exploring the mechanism for the climate transition in the southern hemispheric
stratosphere around 2000.

**Acknowledgements:**
This work is supported by the National Natural Science Foundation of China
(41630421 and 42075062). We thank Institute Pierre Simon Laplace (IPSL), NCEP and
NCAR, National Aeronautics and Space Administration (NASA) and Japan
Meteorological Agency (JMA) for providing ERA-Interim, NCEP-2, MERRA-2 and
JRA-55 datasets. We thank National Oceanic and Atmospheric Administration (NOAA)
for providing ERSST v5 dataset and IPO index. We also thank the scientific team at
NCAR for providing CESM-1 model. Finally, we thank the computing support
provided by the College of Atmospheric Sciences, Lanzhou University.

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

| Experiments | Descriptions |
|---|---|
| sstctrl | Control run. Seasonal cycle of monthly mean global SST data over 1980-2000 is derived from the ERSST v5 dataset. Fixed values of ozone greenhouse gases and aerosol fields in 2000 are used. |
| sstNH | As in sstctrl, but with linear increments of SST in September over 2000-2017 in NH (20°N-70°N). The applied global SST anomalies are shown in Fig. 7a. |
| sstSH | As in sstctrl, but with linear increments of SST in September over 2000-2017 in SH (20°S-70°S). The applied global SST anomalies are shown in Fig. 7b. |
| ssttrop | As in sstctrl, but with linear increments of SST in September over 2000-2017 in the tropics (20°S-20°N). The applied global SST anomalies are shown in Fig. 7c. |
| sstSHtrop | As in sstctrl, but with linear increments of SST in September over 2000-2017 in SHtrop (20°N-70°S). The applied global SST anomalies are shown in Fig. 7d. |

707 **Table 2.** Configurations of experiments for the ozone recovery trend.

| Experiments | Descriptions |
|---|---|





| O3ctrl | Control run. The seasonal cycle of monthly averaged global SST data over 1980-2000 is derived from ERSST v5 dataset. The seasonal cycle of monthly mean three-dimensional global ozone over 1980-2000 is derived from MERRA-2 dataset. The GHGs and aerosol fields are specified to be fixed values in 2000. |
|---|---|
| O3sen | As in O3ctrl, but superposed with linear increments of global ozone in September over 2001-2017. The ozone data in 2002 are removed when the linear increments are calculated. The applied ozone anomalies in Southern Hemisphere are shown in Fig. S5. |

**Table 3.** Correlations of stratospheric vertical wave flux time series (area-weighted
from 100 hPa to 30 hPa over 70°S-50°S) between different reanalysis dataset.

| | ERA-Interim | JRA-55 | MERRA-2 | NCEP-2 |
|---|---|---|---|---|
| ERA-Interim | 1.00 (p=0.00) | 0.99 (p<0.01) | 0.98 (p<0.01) | 0.93 (p<0.01) |
| JRA-55 | | 1.00 (p=0.00) | 0.98 (p<0.01) | 0.93 (p<0.01) |
| MERRA-2 | | | 1.00 (p=0.00) | 0.94 (p<0.01) |
| NCEP-2 | | | | 1.00 (p=0.00) |


**Figure captions:**
**FIG. 1.** Trends of southern hemisphere (a, d) stratospheric E-P flux (arrows, units of
horizontal and vertical components are $10^5$ and $10^3$ kg·s$^{-2}$ per year, respectively) and its
divergence (shadings) with their (b, e) wave-1 components and (c, f) wave-2
components over (a, b, c) 1980-2000 and (d, e, f) 2000-2017 in September derived from
MERRA-2 dataset. The stippled regions indicate the trend of E-P flux divergence
significant at/above the 90% confidence level. The green contours from outside to
inside (corresponding to p=0.1, 0.05) indicate the trend of vertical E-P flux significant





at the 90% and 95% confidence level, respectively.
**FIG. 2.** (a) The mean time series (solid line) and piecewise (during 1980-2000 and
2000-2018) linear regressions (dashed lines) of vertical E-P flux area-weighted from
100 hPa to 30 hPa over 70°S-50°S in September during 1980-2018 derived from ERA-
Interim (yellow), MERRA-2 (blue), JRA-55 (red) and NCEP-2 (green). Figure (b) is
the same as Figure (a), except for that the data in 2002 are removed. (c, d, e, f) The
trends (dots) and uncertainties (error bars) calculated during various periods using the
change-point method with different beginning and ending years (titles). Circles and
squares in Figures (c, d, e, f) represent positive trends from beginning years to change-
point years (x-axes) and negative trends from change-point years to ending years,
respectively. Different colors of dots and error bars in Figures (c, d, e, f) correspond to
colors in Figure (a), which represent trends and uncertainties derived from different
datasets. The long and short error bars in same color reflect the 95% and 90%
confidence intervals calculated by two-tailed t test. The error bar is omitted when the
significance of trend is lower than corresponding confidence level. Negative trends and
corresponding uncertainties with the beginning change-point years after 2005 are also
omitted, since the trend value shows large fluctuation with shortening of time series.
Figures (g, h, i, j) are the same as Figures (c, d, e, f), except that the data in 2002 are
removed when calculating trends and uncertainties.
**FIG. 3.** Trends (shadings) and climatological distributions (contours with an interval
of 20 gpm, positive and negative values are depicted by solid and dashed lines
respectively, zeroes are depicted by thick solid lines) of southern hemispheric (a) 500





hPa geopotential height zonal deviations with their (b) wave-1 component and (c)
wave-2 component in September during 2000−2017 derived from MERRA-2 dataset.
Trends of southern hemispheric (d) tropospheric E-P flux (arrows, units of horizontal
and vertical components are $3\times10^5$ and $3\times10^3$ kg s$^{-2}$ per year, respectively) and its
vertical component (shading) with their (e) wave-1 component and (f) wave-2
component in September during 2000−2017 derived from MERRA-2 dataset. The
stippled regions represent the trend significant at/above the 90% confidence level.
**FIG. 4.** Trends of SST in September over (a) 1980-2000 and (b) 2000-2017 derived
from ERSST v5 dataset. The stippled regions represent the trends significant at/above
the 90% confidence level.
**FIG. 5.** Trend significance of the first three SST principal components (PCs) in (a) the
extratropical southern hemisphere (SH, 70°S-20°S), (b) the tropics (TROP, 20°S-20°N),
(c) the extratropical northern hemisphere (NH, 20°N-70°N), (d) the extratropical
southern hemisphere and the tropics (SHtrop, 70°S-20°N), (e) the extratropical northern
hemisphere and the tropics (NHtrop, 20°S-70°N), (f) the globe (70°S-70°N) and the
corresponding (g, h, i, j, k, l) correlation significances between them and vertical E-P
flux (Fz, area-weighted from 100 hPa to 30 hPa over 70°S-50°S) during different
beginning years (x-axes) and ending years (y-axes). The red and blue dots indicate
positive and negative trend or correlation coefficient are significant, respectively. The
black dots indicate the trends or correlation coefficients are not significant. The stars
indicate that the trends and the corresponding correlation coefficients are both
significant. Each panel is divided into three regions from bottom to top, corresponding





to distinguish whether the trends and correlations are significant or not is the 90%
confidence level.
**FIG. 6.** (a, b, c) The first three EOF patterns of SST in SHtrop region. (d, e, f) The
original time series of the first three principle components (PCs, blue solid lines
correspond to left inverted y-axes) and stratospheric vertical E-P flux (Fz, area-
weighted from 100 hPa to 30 hPa over 70°S-50°S, red solid lines correspond to right y-
axes) in September during 2000-2017. The blue and red dashed lines in (d, e, f)
represent the linear regressions of PC time series and Fz time series, respectively. The
meaning of (g, h, i) are the same as (d, e, f) correspondingly, except the detrended time
series. The unbracketed and bracketed numbers in (g, h, i) represent the correlation
coefficients between detrended PC time series and Fz time series and the corresponding
p values calculated by two-tailed t test, respectively.
**FIG. 7.** Differences in SST forcing field between sensitive experiments ((a) sstNH; (b)
sstSH; (c) ssttrop; (d) sstSHtrop) and the control experiment (sstctrl).
**FIG. 8.** Differences (shadings) of (a, d, g, j) 500 hPa geopotential height zonal
deviations with their (b, e, h, k) wave-1 component and (c, f, i, l) wave-2 component
between sensitive experiments ((a, b, c) sstNH; (d, e, f) sstSH; (g, h, i) ssttrop; (j, k, l)
sstSHtrop) and the control experiment (sstctrl). The mean distributions (contours with
an interval of 20 gpm, positive and negative values are depicted by solid and dashed
lines respectively, zeroes are depicted by thick solid lines) of them are derived from the
control experiment. The stippled regions represent the mean difference significant





**FIG. 9.** Differences of (a, d, g, j) stratospheric E-P flux (arrows, units in horizontal and vertical components are $0.05\times10^7$ and $0.05\times10^5$ kg·s$^{-2}$, respectively) and its divergence (shadings) with their (b, e, h, k) wave-1 component and (c, f, i, l) wave-2 component between sensitive experiments ((a, b, c) sstNH; (d, e, f) sstSH; (g, h, i) ssttrop; (j, k, l) sstSHtrop) and the control experiment (sstctrl). The stippled regions represent the mean differences of E-P flux divergence significant at/above the 90% confidence level. The green contours from outside to inside (corresponding to p=0.1, 0.05) represent the mean differences of vertical E-P flux significant at the 90% and 95% confidence levels, respectively.

**FIG. 10.** (a, b, c, d) Frequency distributions (pillars, blue for control experiment and orange for sensitive experiments) of vertical E-P flux (Fz, area-weighted from 200 hPa to 10 hPa over 70°S-50°S) and its 5-point low-pass filtered fitting curves (solid lines, blue for control experiment and red for sensitive experiments) derived from 100 ensemble members of the control experiment (sstctrl) and sensitive experiments ((a) sstNH; (b) sstSH; (c) ssttrop; (d) sstSHtrop), respectively. (e) Mean differences (grey pillars) and corresponding uncertainties (error bars) of Fz between sensitive experiments and the control experiment. The blue and red error bars reflect the 90% and 95% confidence levels calculated by two-tailed t test, respectively. The error bar is omitted when the significance of mean difference is lower than the corresponding confidence level.

**FIG. 11.** Differences of (a) stratospheric E-P flux (arrows, units in horizontal and



vertical components are $0.02\times10^7$ and $0.05\times10^5$ kg·s$^{-2}$, respectively) and its divergence
(shadings) with their (b) wave-1 component and (c) wave-2 component between the
sensitive experiment (O3sen) and the control experiment (O3ctrl). The stippled regions
represent the mean differences of E-P flux divergence significant at/above the 90%
confidence level. The green contours from outside to inside (corresponding to p=0.1,
0.05) represent the mean differences of vertical E-P flux significant at the 90% and 95%
confidence levels, respectively. (d) Frequency distributions (pillars, blue for O3ctrl and
orange for O3sen) of vertical E-P flux (Fz, area-weighted from 200 hPa to 10 hPa over
70°S-50°S) and it 5-point low-pass filtered fitting curves (solid lines, blue for O3ctrl
and red for O3sen) derived from 100 ensemble members.
**FIG. 12.** (a) Trends of southern hemispheric Brewer-Dobson circulation (arrows, units
in horizontal and vertical components are $0.2\times10^{-2}$ and $0.2\times10^{-4}$ m·s$^{-1}$ per year,
respectively) and its stream function (shadings) in September during (a) 1980-2000 and
(b) 2000-2017 derived from MERRA-2 dataset. Data in 2002 are removed when trends
are calculated in Figure (b). (c) Differences of Brewer-Dobson circulation (arrows,
units in horizontal and vertical components are $10^{-2}$ and $10^{-4}$ m·s$^{-1}$, respectively) and its
stream function (shadings) between the O3ctrl and O3sen. (d, e, f) Differences of
Brewer-Dobson circulation and its stream function between the control experiment
(sstctrl) and various sensitive experiments ((d) sstSH; (e) ssttrop; (f) sstSHtrop) with
SST changes. The stippled regions represent the trends or differences of the stream
function significant at/above the 90% confidence level. The green contours from
outside to inside (corresponding to p=0.1, 0.05) represent the trends or differences of





the vertical components significant at the 90% and 95% confidence levels, respectively.

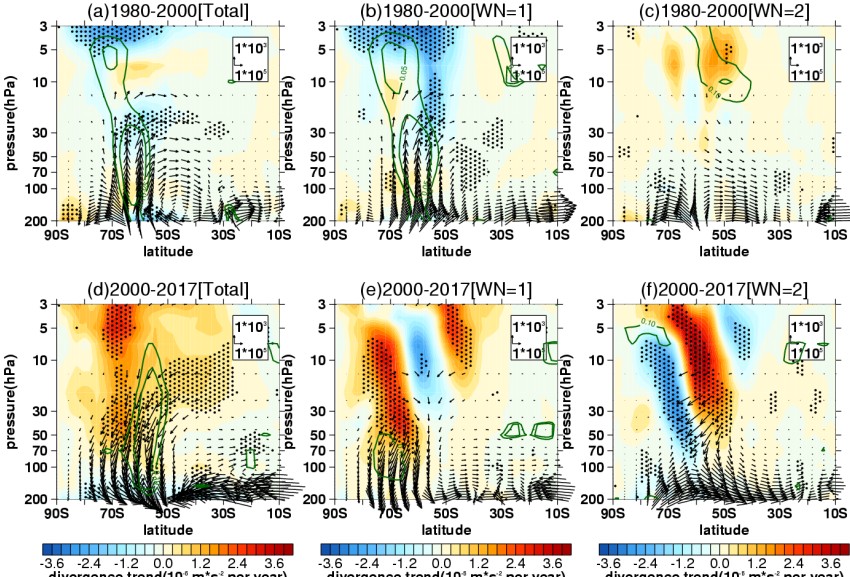


**FIG. 1.** Trends of southern hemisphere (a, d) stratospheric E-P flux (arrows, units of
horizontal and vertical components are $10^5$ and $10^3$ kg·s$^{-2}$ per year, respectively) and its
divergence (shadings) with their (b, e) wave-1 components and (c, f) wave-2
components over (a, b, c) 1980-2000 and (d, e, f) 2000-2017 in September derived from
MERRA-2 dataset. The stippled regions indicate the trend of E-P flux divergence
significant at/above the 90% confidence level. The green contours from outside to
inside (corresponding to p=0.1, 0.05) indicate the trend of vertical E-P flux significant
at the 90% and 95% confidence level, respectively.



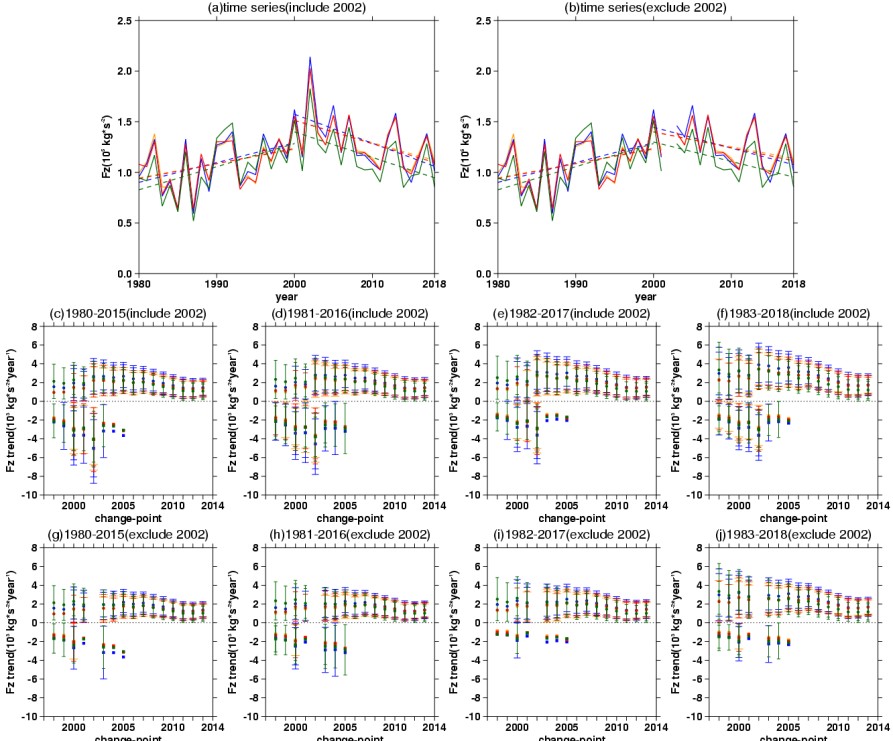

**FIG. 2.** (a) The mean time series (solid line) and piecewise (during 1980-2000 and

2000-2018) linear regressions (dashed lines) of vertical E-P flux area-weighted from

100 hPa to 30 hPa over 70°S-50°S in September during 1980-2018 derived from ERA-

Interim (yellow), MERRA-2 (blue), JRA-55 (red) and NCEP-2 (green). Figure (b) is

the same as Figure (a), except for that the data in 2002 are removed. (c, d, e, f) The

trends (dots) and uncertainties (error bars) calculated during various periods using the

change-point method with different beginning and ending years (titles). Circles and

squares in Figures (c, d, e, f) represent positive trends from beginning years to change-

point years (x-axes) and negative trends from change-point years to ending years,

respectively. Different colors of dots and error bars in Figures (c, d, e, f) correspond to

colors in Figure (a), which represent trends and uncertainties derived from different





datasets. The long and short error bars in same color reflect the 95% and 90%
confidence intervals calculated by two-tailed t test. The error bar is omitted when the
significance of trend is lower than corresponding confidence level. Negative trends and
corresponding uncertainties with the beginning change-point years after 2005 are also
omitted, since the trend value shows large fluctuation with shortening of time series.
Figures (g, h, i, j) are the same as Figures (c, d, e, f), except that the data in 2002 are
removed when calculating trends and uncertainties.

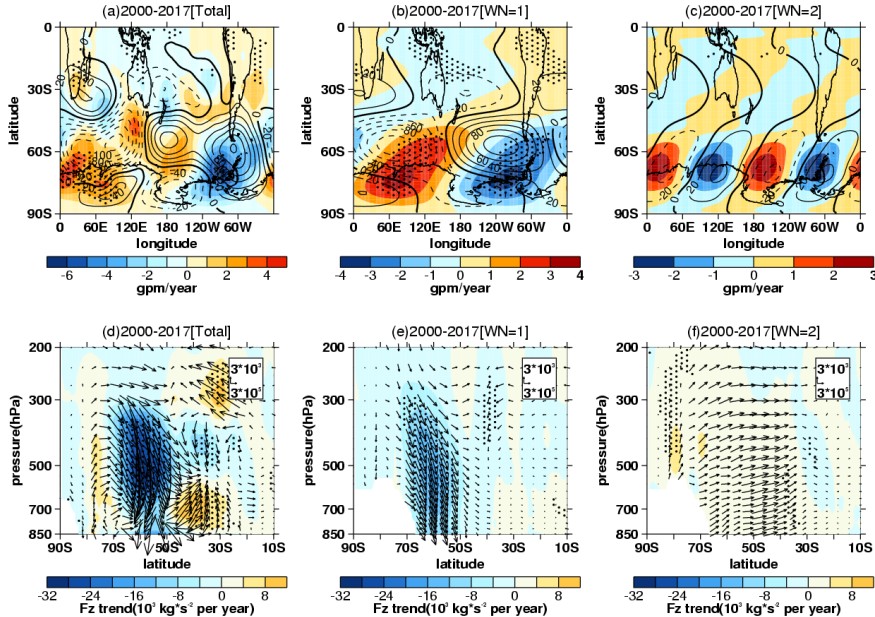


**FIG. 3.** Trends (shadings) and climatological distributions (contours with an interval
of 20 gpm, positive and negative values are depicted by solid and dashed lines
respectively, zeroes are depicted by thick solid lines) of southern hemispheric (a) 500
hPa geopotential height zonal deviations with their (b) wave-1 component and (c)
wave-2 component in September during 2000−2017 derived from MERRA-2 dataset.
Trends of southern hemispheric (d) tropospheric E-P flux (arrows, units of horizontal



and vertical components are $3\times10^5$ and $3\times10^3$ kg s$^{-2}$ per year, respectively) and its
vertical component (shading) with their (e) wave-1 component and (f) wave-2
component in September during 2000−2017 derived from MERRA-2 dataset. The
stippled regions represent the trend significant at/above the 90% confidence level.

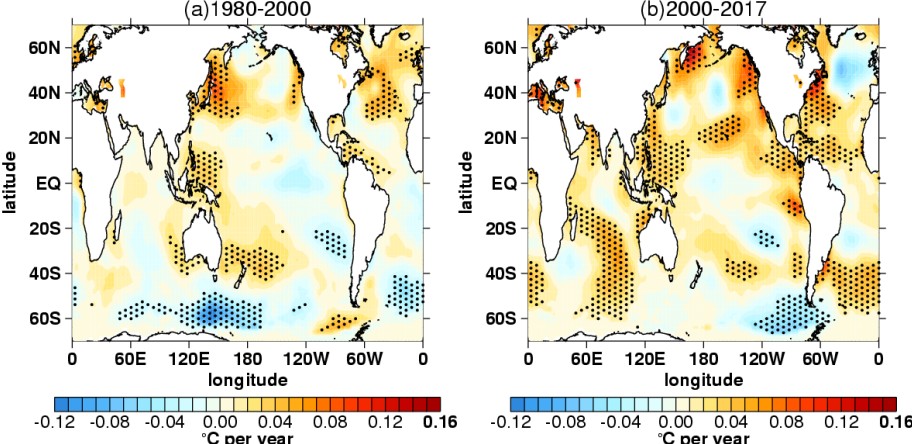


**FIG. 4.** Trends of SST in September over (a) 1980-2000 and (b) 2000-2017 derived
from ERSST v5 dataset. The stippled regions represent the trends significant at/above
the 90% confidence level.

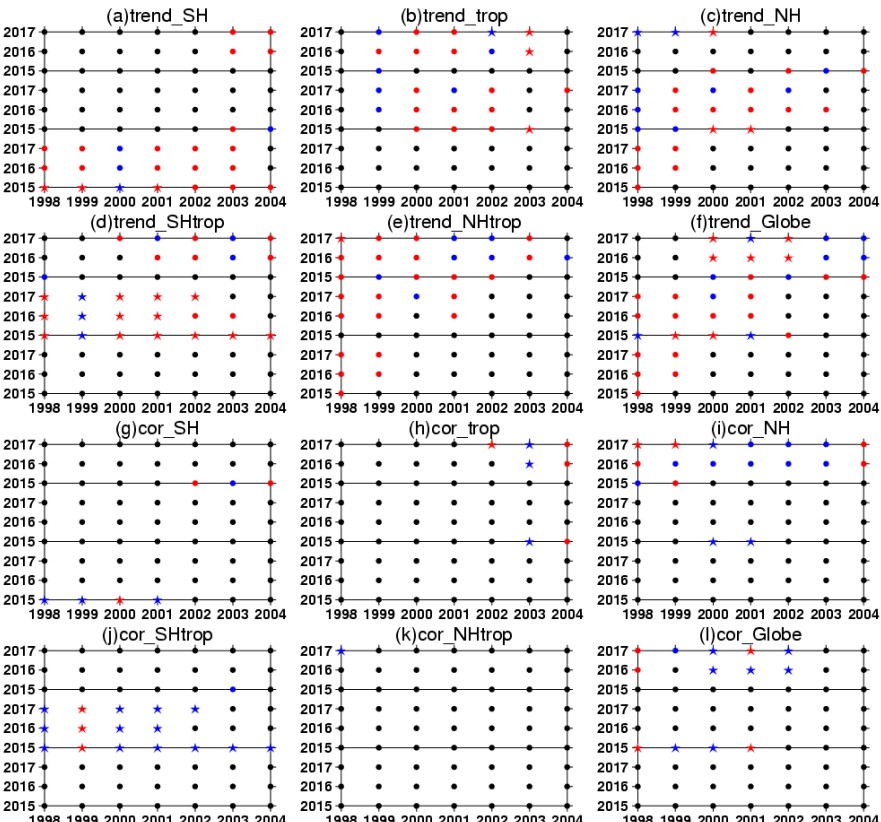


**FIG. 5.** Trend significance of the first three SST principal components (PCs) in (a) the

extratropical southern hemisphere (SH, 70°S-20°S), (b) the tropics (TROP, 20°S-20°N),

(c) the extratropical northern hemisphere (NH, 20°N-70°N), (d) the extratropical

southern hemisphere and the tropics (SHtrop, 70°S-20°N), (e) the extratropical northern

hemisphere and the tropics (NHtrop, 20°S-70°N), (f) the globe (70°S-70°N) and the

corresponding (g, h, i, j, k, l) correlation significances between them and vertical E-P

flux (Fz, area-weighted from 100 hPa to 30 hPa over 70°S-50°S) during different

beginning years (x-axes) and ending years (y-axes). The red and blue dots indicate

positive and negative trend or correlation coefficient are significant, respectively. The

black dots indicate the trends or correlation coefficients are not significant. The stars

885 indicate that the trends and the corresponding correlation coefficients are both

886 significant. Each panel is divided into three regions from bottom to top, corresponding

887 to the first, the second and the third principal components, respectively. The criterion

888 to distinguish whether the trends and correlations are significant or not is the 90%

889 confidence level.

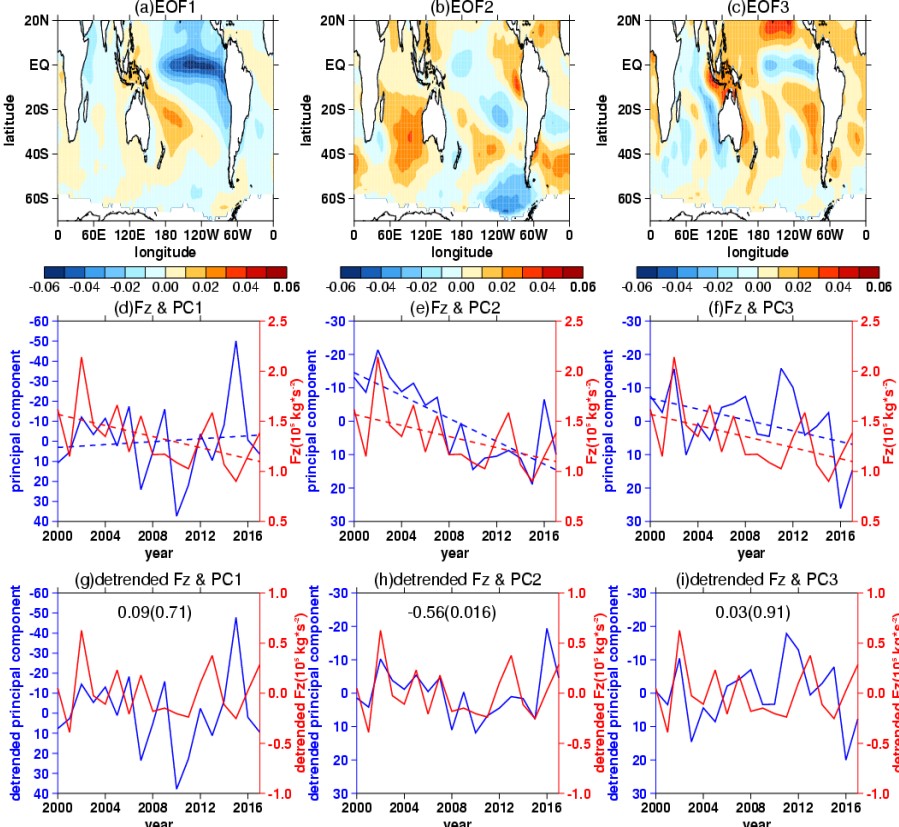

890

891 **FIG. 6.** (a, b, c) The first three EOF patterns of SST in SHtrop region. (d, e, f) The

892 original time series of the first three principle components (PCs, blue solid lines

893 correspond to left inverted y-axes) and stratospheric vertical E-P flux (Fz, area-

894 weighted from 100 hPa to 30 hPa over 70°S-50°S, red solid lines correspond to right y-

895 axes) in September during 2000-2017. The blue and red dashed lines in (d, e, f)





represent the linear regressions of PC time series and Fz time series, respectively. The
meaning of (g, h, i) are the same as (d, e, f) correspondingly, except the detrended time
series. The unbracketed and bracketed numbers in (g, h, i) represent the correlation
coefficients between detrended PC time series and Fz time series and the corresponding
p values calculated by two-tailed t test, respectively.

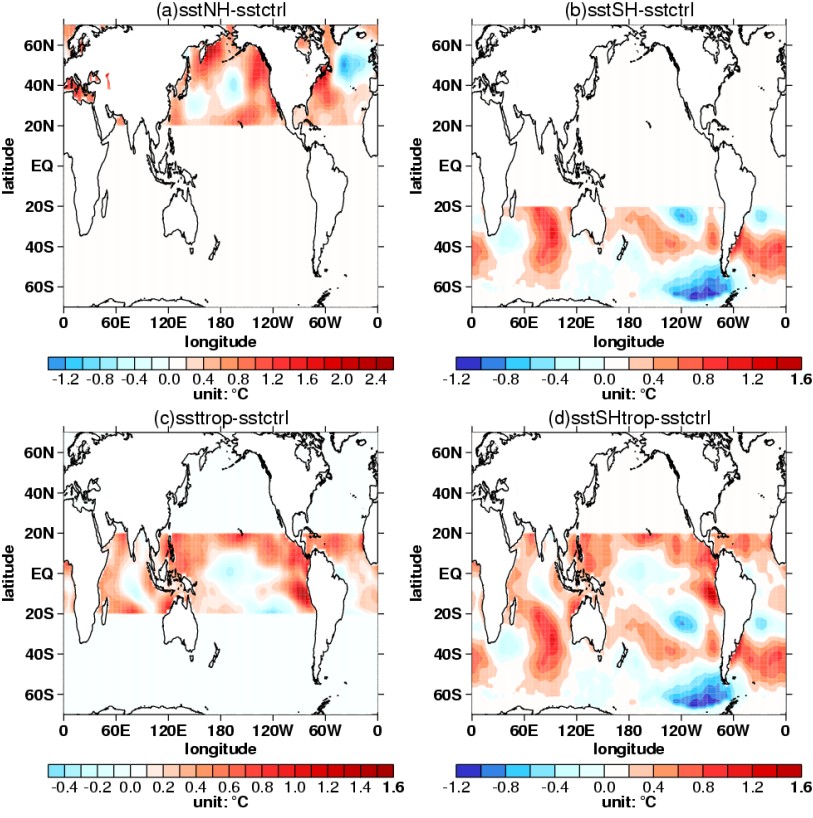

**FIG. 7.** Differences in SST forcing field between sensitive experiments ((a) sstNH; (b)
sstSH; (c) ssttrop; (d) sstSHtrop) and the control experiment (sstctrl).

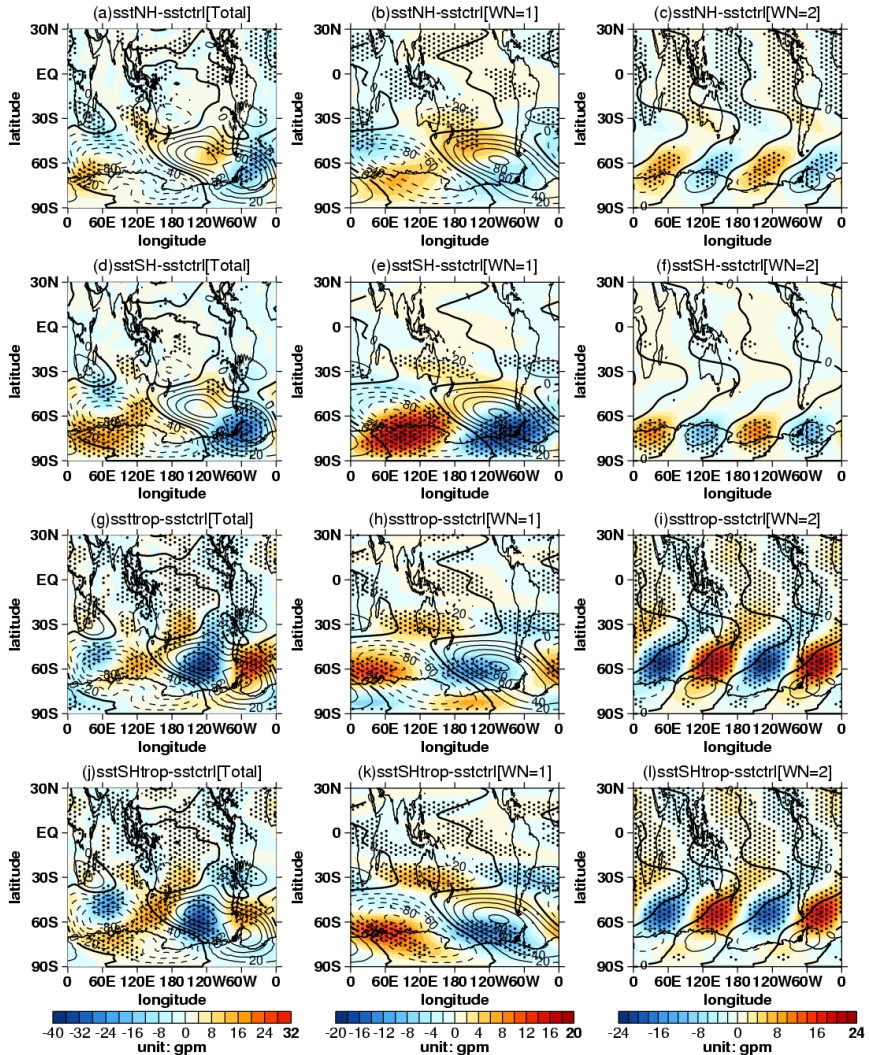

**FIG. 8.** Differences (shadings) of (a, d, g, j) 500 hPa geopotential height zonal

deviations with their (b, e, h, k) wave-1 component and (c, f, i, l) wave-2 component

between sensitive experiments ((a, b, c) sstNH; (d, e, f) sstSH; (g, h, i) ssttrop; (j, k, l)

sstSHtrop) and the control experiment (sstctrl). The mean distributions (contours with

an interval of 20 gpm, positive and negative values are depicted by solid and dashed

lines respectively, zeroes are depicted by thick solid lines) of them are derived from the

control experiment. The stippled regions represent the mean difference significant
at/above the 90% confidence level.

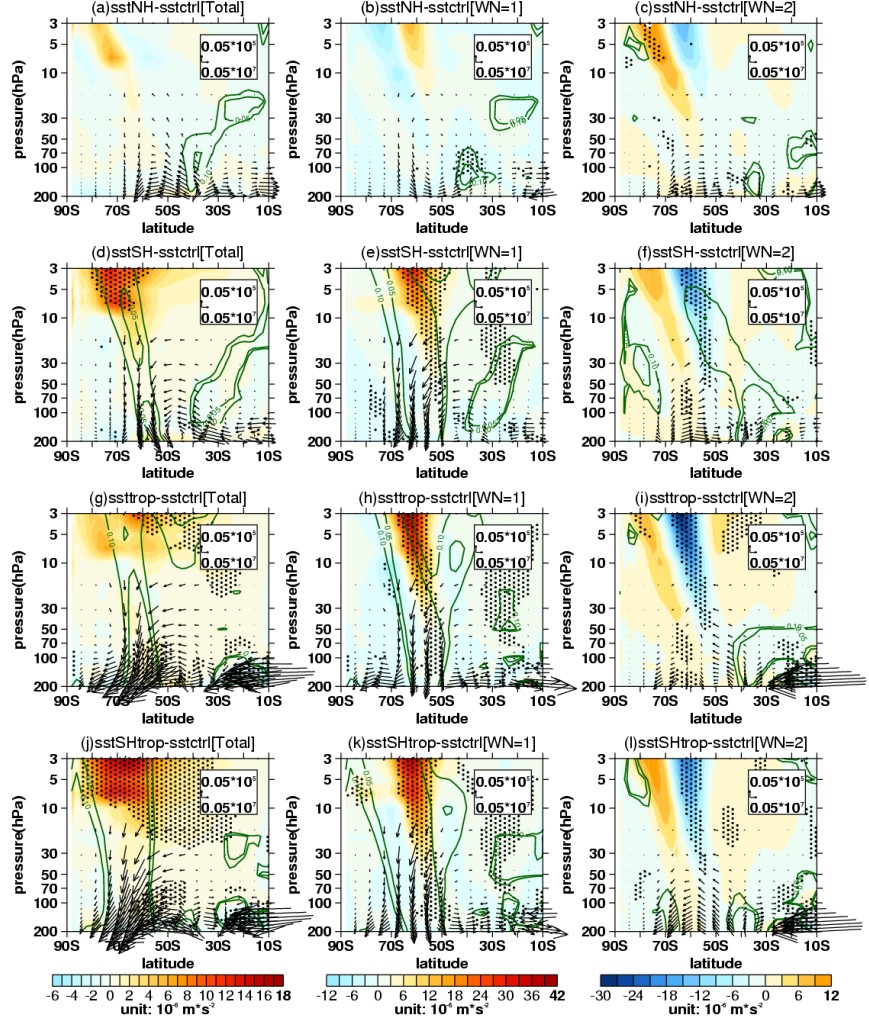

**FIG. 9.** Differences of (a, d, g, j) stratospheric E-P flux (arrows, units in horizontal and
vertical components are $0.05\times10^7$ and $0.05\times10^5$ kg·s$^{-2}$, respectively) and its divergence
(shadings) with their (b, e, h, k) wave-1 component and (c, f, i, l) wave-2 component
between sensitive experiments ((a, b, c) sstNH; (d, e, f) sstSH; (g, h, i) ssttrop; (j, k, l)
sstSHtrop) and the control experiment (sstctrl). The stippled regions represent the mean
differences of E-P flux divergence significant at/above the 90% confidence level. The
green contours from outside to inside (corresponding to p=0.1, 0.05) represent the mean
differences of vertical E-P flux significant at the 90% and 95% confidence levels,
respectively.

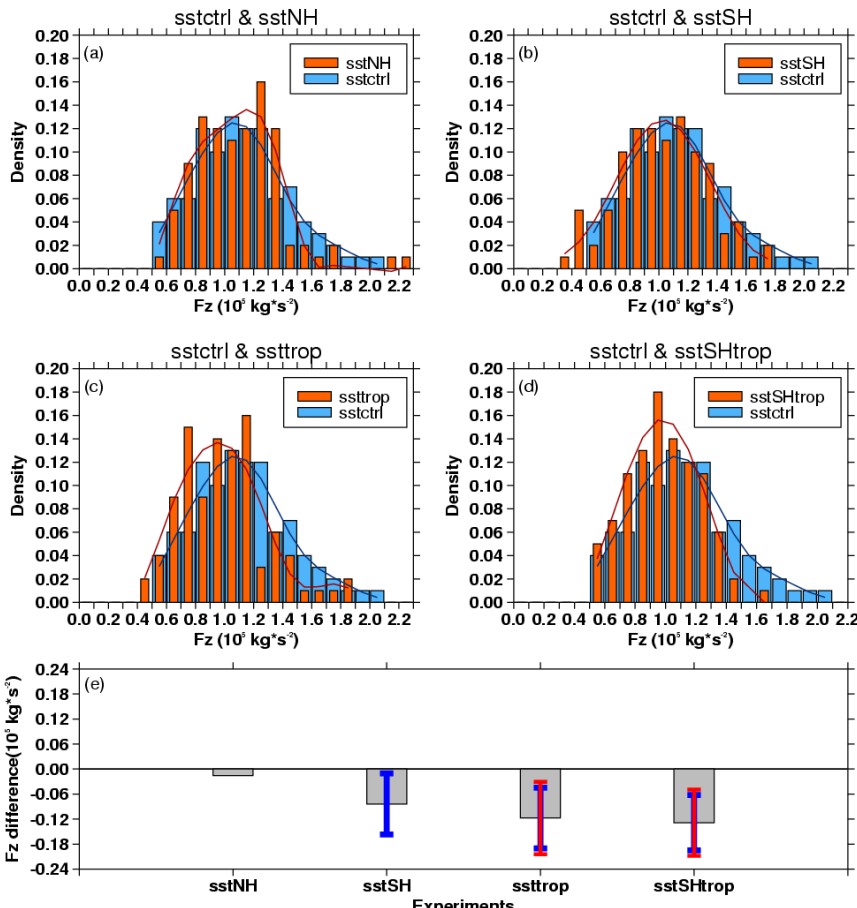


**FIG. 10.** (a, b, c, d) Frequency distributions (pillars, blue for control experiment and
orange for sensitive experiments) of vertical E-P flux (Fz, area-weighted from 200 hPa
to 10 hPa over 70°S-50°S) and its 5-point low-pass filtered fitting curves (solid lines,
blue for control experiment and red for sensitive experiments) derived from 100



ensemble members of the control experiment (sstctrl) and sensitive experiments ((a)
sstNH; (b) sstSH; (c) ssttrop; (d) sstSHtrop), respectively. (e) Mean differences (grey
pillars) and corresponding uncertainties (error bars) of Fz between sensitive
experiments and the control experiment. The blue and red error bars reflect the 90%
and 95% confidence levels calculated by two-tailed t test, respectively. The error bar is
omitted when the significance of mean difference is lower than the corresponding
confidence level.

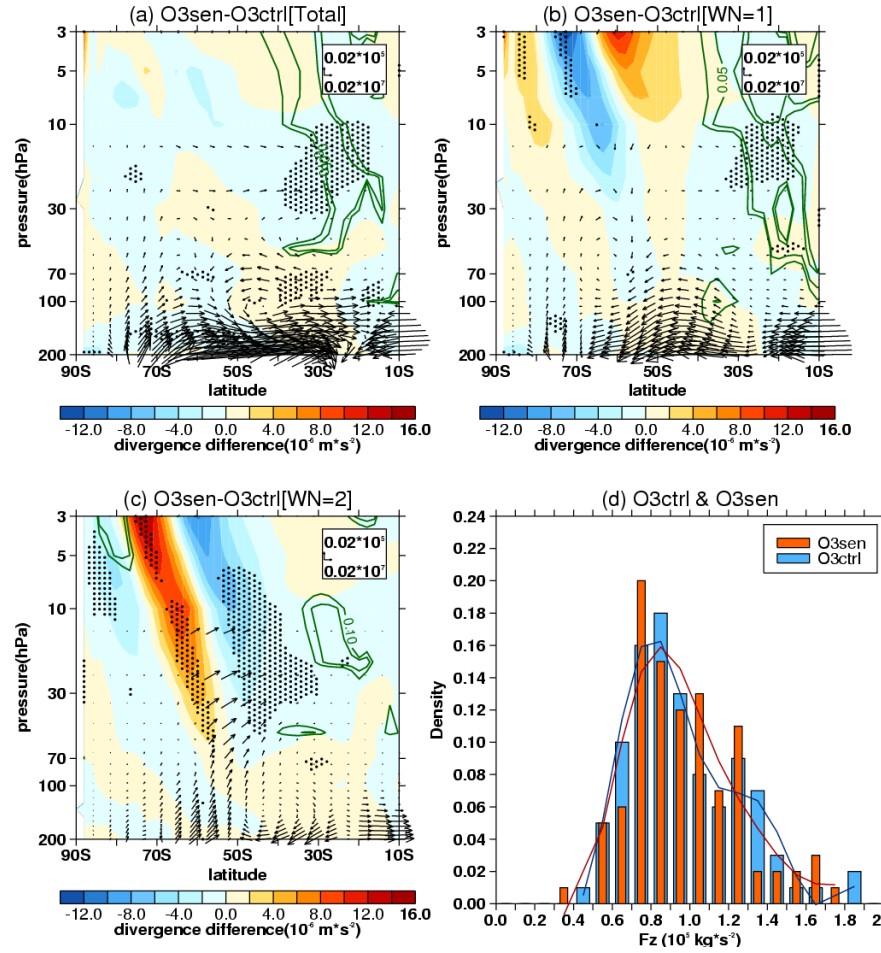

**FIG. 11.** Differences of (a) stratospheric E-P flux (arrows, units in horizontal and





vertical components are $0.02\times10^7$ and $0.05\times10^5$ kg·s$^{-2}$, respectively) and its divergence
(shadings) with their (b) wave-1 component and (c) wave-2 component between the
sensitive experiment (O3sen) and the control experiment (O3ctrl). The stippled regions
represent the mean differences of E-P flux divergence significant at/above the 90%
confidence level. The green contours from outside to inside (corresponding to p=0.1,
0.05) represent the mean differences of vertical E-P flux significant at the 90% and 95%
confidence levels, respectively. (d) Frequency distributions (pillars, blue for O3ctrl and
orange for O3sen) of vertical E-P flux (Fz, area-weighted from 200 hPa to 10 hPa over
70°S-50°S) and it 5-point low-pass filtered fitting curves (solid lines, blue for O3ctrl
and red for O3sen) derived from 100 ensemble members.

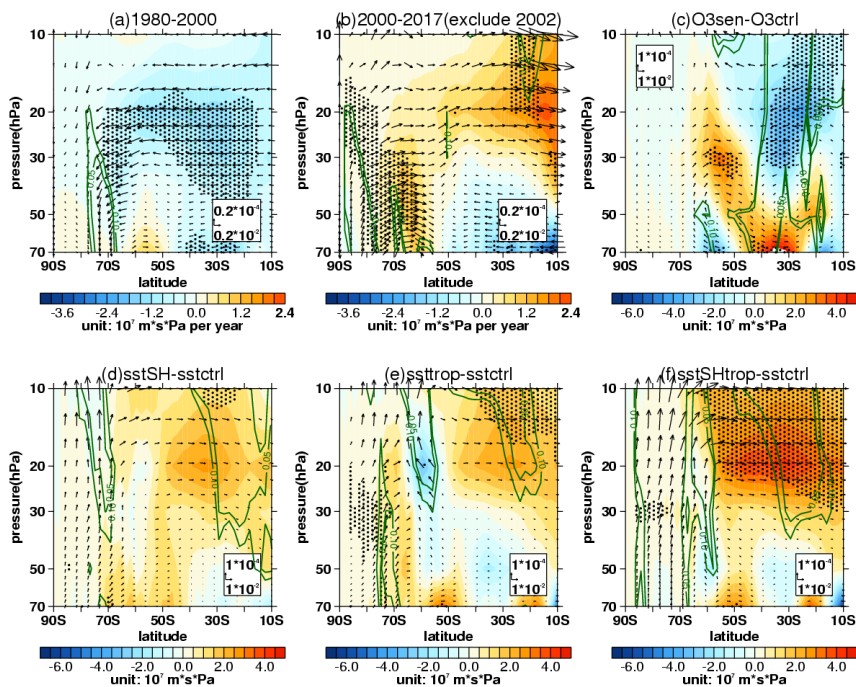


**FIG. 12.** (a) Trends of southern hemispheric Brewer-Dobson circulation (arrows, units
in horizontal and vertical components are $0.2\times10^{-2}$ and $0.2\times10^{-4}$ m·s$^{-1}$ per year,



respectively) and its stream function (shadings) in September during (a) 1980-2000 and
(b) 2000-2017 derived from MERRA-2 dataset. Data in 2002 are removed when trends
are calculated in Figure (b). (c) Differences of Brewer-Dobson circulation (arrows,
units in horizontal and vertical components are $10^{-2}$ and $10^{-4}$ m·s$^{-1}$, respectively) and its
stream function (shadings) between the O3ctrl and O3sen. (d, e, f) Differences of
Brewer-Dobson circulation and its stream function between the control experiment
(sstctrl) and various sensitive experiments ((d) sstSH; (e) ssttrop; (f) sstSHtrop) with
SST changes. The stippled regions represent the trends or differences of the stream
function significant at/above the 90% confidence level. The green contours from
outside to inside (corresponding to p=0.1, 0.05) represent the trends or differences of
the vertical components significant at the 90% and 95% confidence levels, respectively.