# Peer review of "Weakening of Antarctic Stratospheric Planetary Wave Activities in Early Austral Spring Since the Early 2000s: A Response to Sea Surface Temperature Trends"

_Atmospheric Chemistry and Physics, 2021_

## Referee Comment (RC2)

Review of "Weakening of Antarctic Stratospheric Planetary Wave Activities in Early Austral Spring Since the Early 2000s: A Response to Sea Surface Temperature Trends" by Hu et al.

The authors present a comprehensive study on the trend in the planetary wave activities in September over the Antarctic stratosphere from 1980 to 2018. Using reanalysis data and numerical simulations, the authors intend to answer two questions: (1) Has the stratospheric planetary wave activity trend in the southern hemisphere been shifting since 2000? (2) What are the factors responsible for the trend shifting. The authors did a good job in address the first question. For the second question, there is a large room for improvement.

The authors stated that the changes in the stratospheric planetary wave activity trend in the southern hemisphere may be related to changes in SST, stratospheric ozone, and IPO. This paper discussed the effect of SST mostly, which was well done. However, I agree with the other reviewer that time-lags should be further considered in the analysis. The effect of stratospheric ozone is poorly addressed. In the abstract and text, it is stated "The responses of stratospheric wave activities in the southern hemisphere to stratospheric ozone recovery is not significant in simulations". This provides no useful information to the reader as it did not answer if there is such an effect. How well does the CESM model simulate the evolution of stratospheric ozone? Specifically, Figs. 2a and 2b show a clear shift of the trend in stratospheric planetary wave activity over the southern hemisphere around 2000. It would be useful to compare it with the time series of both SST and stratospheric ozone, and both are missing in this paper. Previous studies have shown an inflection point around 1998 in the time series of stratospheric ozone from 1980-2018. An inflection point may not be very apparent in the SST time series. These may provide the authors with some hints for further analysis.

The authors can revise their paper considering the above-mentioned points.

Specific:
L94, "…SST trend". Change to "…changes in SST".

L118, "BDC" is defined in the abstract. Should it be defined in the text?

L148, what is H?

L161, add an "a" before the first "zonal", and a "the" before the second "zonal".

L180-188, provide some references.

L372, "…the SST trends". Change to "…the change in SST".

Be consistent. Equ. or Equation?

Figs. 2a and 2b. leave a space before and after "time series" in the title.

References:
The references are not arranged exactly in the alphabetical order by the author's name.

---

## Author Comment (AC1)

**Responses to Referee's Comments**

**Weakening of Antarctic Stratospheric Planetary in Early Austral Spring Since the Early 2000s: A Response to Sea Surface Temperature Trends**
**(ACP-2021-395)**

Yihang Hu, Wenshou Tian, Jiankai Zhang, Tao Wang and Mian Xu

September 2021

**Responses to Referee 1**

*The manuscript "Weakening of Antarctic Stratospheric Planetary Wave Activities in Early Austral Spring Since the Early 2000s: A Response to Sea Surface Temperature Trends" by Drs. Hu et al. identified a decreasing trend in the September Antarctic stratospheric wave activity since early 2000s and attribute it to the SST trends in the tropics and southern hemisphere. The modelling evidence presented in the manuscript well supports their conclusions and the paper is logically organized. It is thus recommended to be considered for publication after addressing the following comments:*

**Response:**

We thank the reviewer for the helpful comments and valuable suggestions. We have revised the manuscript carefully according to the reviewer's comments and the detailed point-by-point responses to those comments are listed below.

**Comment #1:**

*First of all, a positive trend in September Antarctic stratospheric wave activity is evident before 2000s (Fig. 2). It would be a more complete study to also simulate the 1980-2000 period similarly to better attribute and contrast the positive trend to the negative trend in the later period, as previous studies (e.g. Hu and Fu 2009) mostly use statistical methods for the attribution. This might require too much time and resource to complete and is thus only a suggestion.*

**Response:**

Thanks for the comment. Hu and Fu (2009) (hereafter HF2009) simulated the trend of eddy heat flux under the observed time-varying SST (Fig. 11b in HF2009), which is relatively weak compared to that derived from reanalysis data (Fig. 6b in HF2009). However, they didn't show the trends of tropospheric wave sources in their paper. Fig. R1 shows the trends of stratospheric wave activities and tropospheric wave sources in September during 1980-2000 derived from MERRA-2 dataset. Note that the trends of stratospheric wave activities are not in accord with the trends of tropospheric wave

sources. The wave-1 component plays a predominate role in increase of stratospheric vertical wave flux during 1980-2000 (Fig. R1a, b). However, the trend of wave-1 component in 500 hPa geopotential height does not show a significant in-phase superposition on its climatology (Fig. R1e). The trend pattern of wave-2 component is in-phase with its climatology (Fig. R1f). These features in Fig. R1 imply that the wave-1 component of tropospheric wave sources was weakening during 1980-2000, while the wave-2 component was strengthening.

We have also conducted an model experiment (sstSHtrop80) forced by the linear increments of September SST during 1980-2000 over tropics and extratropical southern hemisphere (20°N-70°S), which is similar to sstSHtrop discussed in the manuscript. The applied SST anomalies are shown in Fig. R2. The responses of stratospheric wave activities and tropospheric wave sources to sstSHtrop80 are shown in Fig. R3. The responses of tropospheric wave sources are analogous to the trends derived from MERRA-2 to some extent. The response of wave-2 component is in-phase with its climatology (Fig. R3f), while the response of wave-1 component is out-of-phase with its climatology (Fig. R3e). The responses of stratospheric wave activities are consistent with the tropospheric wave sources. The response in wave-1 component is not significant (Fig. R3b), while the wave-2 component plays a dominate role in increase of stratospheric vertical wave flux in the model simulation (Fig. R3a, c). The results from sstSHtrop80 suggest that the SST increases over 20°N-70°S induce a strengthening of September stratospheric wave activity during 1980-2000. But it cannot explain the intensified wave-1 component of stratospheric wave activity shown in Fig. R1b.

Indeed, as the reviewer pointed out, a detailed attribution of the trend of Antarctic stratospheric wave activity during 1980-2000 needs much more efforts. In this manuscript, we mainly focuses on September wave activity after 2000, and some brief discussion about the the trend of Antarctic stratospheric wave activity before 2000 is added in the revised manuscript.

[Figure]

**FIG. R1.** (a-c) Trends of southern hemisphere (a) stratospheric E-P flux (arrows, units of horizontal and vertical components are $10^5$ and $10^3$ kg·s$^{-2}$ per year, respectively) and its divergence (shadings) with their (b) wave-1 components and (c) wave-2 components over 1980-2000 in September derived from MERRA-2 dataset. (d-f) Trends (shadings) and climatological distributions (contours with an interval of 20 gpm, positive and negative values are depicted by solid and dashed lines respectively, zeroes are depicted by thick solid lines) of southern hemispheric (d) 500 hPa geopotential height zonal deviations with their (e) wave-1 component and (f) wave-2 component in September during 1980-2000 derived from MERRA-2 dataset. The stippled regions indicate the trends of E-P flux divergence or geopotential height significant at/above the 90% confidence level. The green contours from outside to inside (corresponding to p=0.1, 0.05) in Figs. R1a-c indicate the trend of vertical E-P flux significant at the 90% and 95% confidence level, respectively.

[Figure]

**FIG. R2.** Differences in SST forcing field between sstSHtrop80 and sstctrl.

[Figure]

**FIG. R3.** Differences of (a) stratospheric E-P flux (arrows, units in horizontal and vertical components are $0.02 \times 10^7$ and $0.02 \times 10^5$ kg·s$^{-2}$, respectively) and its divergence (shadings) with their (b) wave-1 component and (c) wave-2 component between

sstSHtrop80 and sstctrl. Differences (shadings) of (d) 500 hPa geopotential height zonal deviations with their (e) wave-1 component and (f) wave-2 component between sstSHtrop80 and sstctrl. The mean distributions (contours with an interval of 20 gpm, positive and negative values are depicted by solid and dashed lines respectively, zeroes are depicted by thick solid lines) of geopotential height zonal deviations are derived from sstctrl. The stippled regions represent the mean differences significant at/above the 90% confidence level. The green contours from outside to inside in Figs. R3a-c represent the mean differences of vertical E-P flux significant at the 90% and 95% confidence levels.

**Reference**

Hu, Y., & Fu, Q.: Stratospheric warming in southern hemisphere high latitudes since 1979, Atmos. Chem. Phys., 9(13), 4329-4340, https://doi.org/10.5194/acp-9-4329-2009, 2009.

**Comment #2:**

*Second, there is usually a time lag for the extratropical atmospheric circulation to respond to the tropical SST anomaly. The authors might need to justify whether it is reasonable to use September SST to drive the extratropical atmospheric circulation for the same month.*

**Response:**

Thanks for the comment. We agree with the reviewer that there is a time lag for the extratropical atmosphere circulation to respond to tropical SST anomalies. As stated in the original manuscript (lines 244-245), the tropical SST anomalies (the linear increments) in experiment ssttrop are also applied in July and August (Fig. R4a, b) to avoid abrupt SST variations from month to month, and the two months are taken as spin-up time. Therefore, whether the SST forcing in July and August also contribute to the weakening of Antarctic stratospheric wave activity in September or not cannot be justified based on the experiment ssttrop only. Here, we performed an additional experiment ssttropAug without September SST anomalies (Fig. R4f) to clarify whether

the weakening of Antarctic stratospheric wave activity is induced by the tropical SST trend at the same month. Like other numerical experiments described in Table 1, the ssttropAug also includes 100 ensemble members that run from July to September forced by the same initial conditions from the 21st year to the 120th year in July generated by free run. The detailed descriptions of ssttropAug and other relevant experiments in the manuscript are displayed together in the Table R1 for comparison. The Figure R4 shows the applied global SST anomalies in ssttrop and ssttropAug from July to September.

The responses of tropospheric wave sources and stratospheric wave activities in ssttropAug are shown in Figs. R5a-c and Figs. R5d-f, respectively. Note that the anomalies of subpolar tropospheric geopotential height in September forced by changes in tropical SST in August does not superpose on their climatological patterns in an evident out-of-phase style (Figs. R5a-c). The anomaly of wave-1 component of geopotential height shows a slight in-phase overlap with its climatology over subpolar region (Fig. R5b). Accordingly, the responses of stratospheric wave activities over subpolar of southern hemisphere are not significant (Figs. R5d-f). The results here suggest that, the decrease of September vertical wave flux induced by SST changes in August is negligible comparing to that in the experiment with anomalous SST forcing in September (Figs. R5g), and the tropical SST trend in September plays a dominate role in weakening of stratospheric wave activity at the same month.

Furthermore, we also use a linear barotropic model (LBM) (e.g. Shaman & Tziperman, 2007; Shaman & Tziperman, 2011) to quantify the time scale for propagation of tropical anomalies to high latitudes. The LBM are developed to solve the barotropic vorticity equation, which is given as Eq. (1):

$$J(\bar{\psi}, \nabla^2\psi') + J(\psi', \nabla^2\bar{\psi} + f) + \alpha\nabla^2\psi' + K\nabla^4\nabla^2\psi' = R \tag{1}$$

where the Jacobian $J(A,B)$ is

$$J(A,B) = \frac{1}{r^2}(\frac{\partial A}{\partial\lambda}\frac{\partial B}{\partial\mu} - \frac{\partial A}{\partial\mu}\frac{\partial B}{\partial\lambda}) \tag{2}$$

the forcing function $R$ is

$$R = -(f + \nabla^2\bar{\psi})D \tag{3}$$

$\psi$ is the streamfunction, $f$ is the Coriolis force, $\alpha$ is the Rayleigh coefficient, $K$ is a diffusion coefficient, $\lambda$ is longitude, $\mu = \sin(\theta)$, $\theta$ is latitude, $r$ is the earth's radius and $D$ is the divergence.

We use the wave-1 component of streamfunction derived from ensemble mean of sstctrl as the background field. In LBM, the initial anomaly is given by the divergence. We set $D = -7.9 \times 10^{-7}$ $s^{-1}$. The divergence forcing field is limited in 40°E-140°W, 10° S-0° (Fig. R6) to ensure the tropical initial anomaly of streamfunction superpose on its background field in an out-of-phase style. The LBM simulated streamfunction anomalies are shown in Figs. R7b-i. Note that the anomalies in tropics only take a few days to arrive the high latitudes in Southern Hemisphere. After about four days, a stable anti-phase superposition of streamfunction is well established in extratropical southern hemisphere (Figs. R7f-i). These results are supported by previous studies (e.g. Shaman & Tziperman, 2011), which also indicate that the horizontal propagation of anomaly in atmosphere takes a few days.

Previous studies also reported that it takes about 4 days for wave-1 to propagate from troposphere into stratosphere and 1-2 days for wave-2 (e.g. Randel, 1987). We agree with the reviewer that the tropical oceans affect the stratosphere at mid-high latitudes with a lag of several days. However, the SST forcing field applied in CESM is on monthly scale. It is reasonable to use September SST trend to drive and explain the trends of extratropical circulation and wave activity at the same month.

**Table R1.** Configurations of sstctrl, ssttrop and ssttropAug.

| Experiments | Descriptions |
| --- | --- |
| sstctrl | Control run. Seasonal cycle of monthly mean global SST data over 1980-2000 is derived from the ERSST v5 dataset. Fixed values of ozone, greenhouse gases and aerosol fields in 2000 are used. |
| ssttrop | As in sstctrl, but with linear increments of SST in September over 2000-2017 superposed on the tropics (20°S-20°N). As shown in |

| | Fig. R1a-c, the global SST anomalies are applied from July to September,. |
|---|---|
| ssttropAug | As in sstctrl, but with linear increments of SST in August over 2000-2017 superposed on the tropics (20°S-20°N). As shown in Fig R1d-f, the SST anomalies are only applied from July to August. |

[Figure]

**FIG. R4.** Differences of SST forcing fields in July (a, d), August (b, e) and September (c, f) between the sensitive experiments ((a, b, c) ssttrop; (d, e, f) ssttropAug) and the control experiment (sstctrl).

[Figure]

**FIG. R5.** (a-c) The responses of tropospheric wave sources in experiment ssttropAug: differences of (a) 500 hPa geopotential height zonal deviations with their (b) wave-1 component and (c) wave-2 component between ssttropAug and sstctrl. The mean distributions (contours with an interval of 20 gpm, positive and negative values are depicted by solid and dashed lines, respectively, zeros are depicted by thick solid lines) of them are derived from sstctrl. (d-f) The responses of stratospheric wave activities in experiment ssttropAug: differences of (d) stratospheric E-P flux (arrows, units in horizontal and vertical components are $0.05 \times 10^7$ and $0.05 \times 10^5$ kg·s$^{-2}$, respectively) and its divergence (shadings) with their (e) wave-1 component and (f) wave-2 component between ssttropAug and sstctrl. The stippled regions in Figs. R5a-f represent the mean difference significant at/above the 90% confidence level. The green contours from outside to inside (corresponding to p=0.1 and 0.05) in Figs. R5d-f represent the mean

differences of vertical E-P flux significant at the 90% and 95% confidence levels, respectively. (g) Mean differences (grey pillars) and corresponding uncertainties (error bars) of Fz (area-weighted from 200 hPa to 10 hPa over 70°S-50°S) between sensitive experiments and the control experiment. The blue and red error bars reflect the 90% and 95% confidence levels calculated by two-tailed t test, respectively.

[Figure]

**FIG. R6.** The initial forcing ( $R = -(f + \nabla^2 \overline{\psi})D$ ) distribution in LBM.

[Figure]

**FIG. R7.** The background field (contours with interval of $10^6$ m²·s⁻¹, positive and negative values are depicted by solid and dashed lines, respectively, zeros are depicted by thick solid lines) of streamfunctions derived from sstctrl and the responses (shading) of streamfunctions derived from (a) ssttrop in CESM and (b-i) the first to eighth model days in LBM.

**Reference**

Randel, W. J.: A study of planetary waves in the southern winter troposphere and stratosphere. Part I: Wave structure and vertical propagation, J. Atmos. Sci., 44(6), 917-935, 1987.

Shaman, J., & Tziperman, E.: Summertime ENSO-North African-Asian Jet teleconnection and implications for the Indian monsoons, Geophys. Res. Lett., 34(11), L11702, https://doi.org/

10.1029/2006GL029143, 2007.

Shaman, J., & Tziperman, E.: An atmospheric teleconnection linking ENSO and southwestern European precipitation, J. Climate., 24(1), 124-139, https://doi.org/10.1175/2010JCLI3590.1, 2011.

**Comment #3:**

*Last, it would be nice to show the simulated EP-flux time series as in Fig. 2 but forced by SSTs to visualize how significant that is compared with the EP-flux trends in reanalysis datasets. Previous studies (e.g. Wang and Waugh, 2012, https://doi.org/10.1029/2011JD017130) found that the extratropical stratospheric wave activity trend is difficult to capture using model simulations with small ensembles. It might help to illustrate the benefit of using large ensembles as in this study.*

**Response:**

Thanks for the comment. The study (hereafter WW2012) mentioned in this comment uses the chemistry-climate model simulations to evaluate the trends of stratospheric temperature, residual circulation as well as wave activity during recent decades (Wang and Waugh, 2012). These simulations are all forced by time varying SSTs, GHGs and ODSs. It's necessary to emphasize that the simulations conducted in our study are all time-slice experiments. We had tried to conduct transient experiments forced by time-varying SST derived from ERSST v5. However, possibly due to the limitation of the model performance, the trends of wave activities we simulated are not significant despite the opposite signs during 1980-2000 and 2000-2018 (Table R2, Fig. R8), which is analogous to Fig. 6 in WW2012. Meanwhile, numerous previous studies also used time-slice experiments to attribute the trends in atmosphere (e.g. Hu et al., 2018; Kang et al., 2011; Zhang et al., 2016) and we perform time-slice experiments mainly for the purpose of attribution rather than generating a real trend. Fig. R9 shows the simulated stratospheric vertical wave flux derived from each ensemble member in our study. Although the scatter plots do not show clear trends like Fig. 2 in manuscript, the differences between ensemble means in values of these dots still indicate the

decrease of stratospheric wave activities induced by SST changes.

**Table R2**. Trends of stratospheric vertical wave flux time series (averaged from 100 hPa to 30 hPa over 70°S-50°S) derived from different transient experiments (tr01, tr02, tr03, tr04, tr05) and ensemble mean of them on piecewise periods (1980-2000 and 2000-2018).

|  | tr01 | tr02 | tr03 | tr04 | tr05 | ensemble mean |
|---|---|---|---|---|---|---|
| 1980-2000 | 0.0091 | -0.012 | 0.012 | 0.0031 | 0.0034 | 0.0031 |
|  | (p=0.44) | (p=0.32) | (p=0.15) | (p=0.67) | (p=0.77) | (p=0.55) |
| 2000-2018 | -0.0060 | 0.0086 | -0.030 | -0.0034 | 0.019 | -0.0023 |
|  | (p=0.61) | (p=0.67) | (p=0.039) | (p=0.87) | (p=0.23) | (p=0.76) |

[Figure]

**FIG. R8.** Time series (solid lines) of vertical E-P flux area-weighted from 100 hPa to 30 hPa over 70°S-50°S in September during 1980-2018 derived from simulations forced by time-varying SST. Five different red solid lines stand for the time series driven by different initial conditions and the black solid line represent the ensemble

mean of them. The straight dashed lines show linear regressions to the ensemble mean on two piecewise period (1980-2000 and 2000-2018).

[Figure]

**FIG. R9.** Stratospheric vertical E-P flux (Fz, area-weighted from 200 hPa to 10 hPa over 70°S-50°S) derived from each ensemble member of control experiment (black squares) and different sensitive experiment (red circles; (a) sstSH; (b) ssttrop; (c) sstSHtrop). Black and red horizontal dashed lines represent the ensemble means derived from control experiment and sensitive experiments, respectively.

**Reference**

Hu, D., Guan, Z., Tian, W., & Ren, R.: Recent strengthening of the stratospheric Arctic vortex

response to warming in the central North Pacific, Nat. Commun., 9(1), 1697. https://doi.org/10.1038/s41467-018-04138-3, 2018.

Kang, S. M. , Polvani, L. M. , Fyfe, J. C. , & Sigmond, M.: Impact of polar ozone depletion on subtropical precipitation, Science, 332(6032), 951-954, https://doi.org/10.1126/science.1202131, 2011.

Wang, L., & Waugh, D., W.: Chemistry-climate model simulations of recent trends in lower stratospheric temperature and stratospheric residual circulation, J. Geophys. Res-Atmos., 117(D9), https://doi.org/10.1029/2011JD017130, 2012.

Zhang, J., Tian, W. , Chipperfield, M. P. , Xie, F. , & Huang, J.: Persistent shift of the arctic polar vortex towards the eurasian continent in recent decades, Nat. Clim. Change. 6, 1094–1099. https://doi.org/10.1038/nclimate3136, 2016.

---

## Author Comment (AC2)

**Responses to Referee's Comments**

**Weakening of Antarctic Stratospheric Planetary in Early Austral Spring Since the Early 2000s: A Response to Sea Surface Temperature Trends**

**(ACP-2021-395)**

Yihang Hu, Wenshou Tian, Jiankai Zhang, Tao Wang and Mian Xu

September 2021

**Responses to Referee 2**

*The authors present a comprehensive study on the trend in the planetary wave activities in September over the Antarctic stratosphere from 1980 to 2018. Using reanalysis data and numerical simulations, the authors intend to answer two questions: (1) Has the stratospheric planetary wave activity trend in the southern hemisphere been shifting since 2000? (2) What are the factors responsible for the trend shifting. The authors did a good job in address the first question. For the second question, there is a large room for improvement.*

**Response:**

We appreciate the reviewer for sparing time to go through the manuscript, providing useful comments and valuable suggestions to improve our manuscript. We have revised the manuscript carefully according to the reviewer's comments and suggestions. The detailed responses are listed as follows:

**General comments:**

*The authors stated that the changes in the stratospheric planetary wave activity trend in the southern hemisphere may be related to change in SST, stratospheric ozone, and IPO. This paper discussed the effect of SST mostly, which was well done. However, I agree with the other reviewer that time-lags should be further considered in the analysis. The effect of stratospheric ozone is poorly addressed. In the abstract and text, it is stated "The responses of stratospheric wave activities in the southern hemisphere to stratospheric ozone recovery is not significant in simulations". This provides no useful information to the reader as it did not answer if there is such an effect. How well does the CESM model simulate the evolution of stratospheric ozone? Specifically, Figs.2a and 2b show a clear shift of the trend in stratospheric planetary wave activity over the southern hemisphere around 2000. It would be useful to compare it with the time series of both SST and stratospheric ozone, and both are missing in this paper. Previous studies have shown an inflection point around 1998 in the time series of*

*stratospheric ozone from 1980-2018. An inflection point may not be very apparent in the SST time series. These may provide the authors with some hints for further analysis.*

**Response:**

Thanks for the comments. The comments include three parts and we will give our responses separately.

**1. Responses to comments about time lags**

The reviewer's first concern is about time lags of the responses, which is the same as the second comment raised by the other reviewer. As the detailed description about numerical experiments has been given in response to the comment #2 from the other reviewer, we just list some main points and display the supporting figures here.

Firstly, we performed an additional experiment ssttropAug in September SST anomalies are excluded to clarify whether the weakening of Antarctic stratospheric wave activity is induced by the tropical SST trend at the same month. The responses of tropospheric wave sources and stratospheric wave activities in ssttropAug are shown in Figs. R1a-c and Figs. R1d-f, respectively. Note that the anomalies of subpolar tropospheric geopotential height in September forced by change in tropical SST in August does not superpose on their climatological patterns in an obvious out-of-phase style (Figs. R1a-c). The anomaly of wave-1 component of geopotential height shows slight in-phase overlap with its climatology over subpolar region (Fig. R1b). Accordingly, the responses of stratospheric wave activities over subpolar of southern hemisphere are not significant (Figs. R1d-f). The decrease of September vertical wave flux induced by SST changes in August is negligible comparing to the experiment includes anomalous SST forcing in September (Figs. R1g), which suggests that the tropical SST trend in September plays a dominate role in weakening of stratospheric wave activity at the same month.

Secondly, we use a linear barotropic model (LBM) (e.g., Shaman & Tziperman, 2007; Shaman & Tziperman, 2011) to quantify the time scale for propagation of tropical anomalies to high latitudes. The LBM simulated streamfunction anomalies are shown in Figs. R2b-i. Note that the anomalies in tropics only take a few days to arrive the high

latitudes in Southern Hemisphere. After about four days, a stable anti-phase superposition of streamfunction is well established in extratropical southern hemisphere (Figs. R2f-i). These results are supported by previous studies (e.g., Shaman & Tziperman, 2011), which also indicate that the horizontal propagation of an anomaly in atmosphere takes a few days.

Previous studies also reported that it takes about 4 days for wave-1 to propagate from troposphere into stratosphere and 1-2 days for wave-2 (e.g. Randel, 1987). We agree with the reviewer that the tropical oceans affect the stratosphere at mid-high latitudes with a lag of several days. However, the SST forcing field applied in CESM is on monthly scale. It is reasonable to use September SST trend to drive and explain the trends of extratropical circulation and wave activity at the same month.

[Figure]

**FIG. R1.** (a-c) The responses of tropospheric wave sources in experiment ssttropAug:

differences of (a) 500 hPa geopotential height zonal deviations with their (b) wave-1 component and (c) wave-2 component between ssttropAug and sstctrl. The mean distributions (contours with an interval of 20 gpm, positive and negative values are depicted by solid and dashed lines, respectively, zeros are depicted by thick solid lines) of them are derived from sstctrl. (d-f) The responses of stratospheric wave activities in experiment ssttropAug: differences of (d) stratospheric E-P flux (arrows, units in horizontal and vertical components are $0.05 \times 10^7$ and $0.05 \times 10^5$ kg·s$^{-2}$, respectively) and its divergence (shadings) with their (e) wave-1 component and (f) wave-2 component between ssttropAug and sstctrl. The stippled regions in Figs. R1a-f represent the mean difference significant at/above the 90% confidence level. The green contours from outside to inside (corresponding to p=0.1 and 0.05) in Figs. R1d-f represent the mean differences of vertical E-P flux significant at the 90% and 95% confidence levels, respectively. (g) Mean differences (grey pillars) and corresponding uncertainties (error bars) of Fz (area-weighted from 200 hPa to 10 hPa over 70°S-50°S) between sensitive experiments and the control experiment. The blue and red error bars reflect the 90% and 95% confidence levels calculated by two-tailed t test, respectively.

[Figure]

**FIG. R2.** The background field (contours with interval of $10^6$ m$^2$·s$^{-1}$, positive and negative values are depicted by solid and dashed lines, respectively, zeros are depicted by thick solid lines) of streamfunction derived from sstctrl and the responses (shading) of streamfunction derived from (a) ssttrop and (b-i) the first to eighth model days in LBM.

**2. Responses to comments about ozone time series and effects of stratospheric ozone**

The reviewer suggests that it is necessary to add ozone time series in the manuscript and give more specific discussions about impacts of ozone recovery on stratospheric wave activity. Following the suggestions, we have added a new section after Section 3 and some text in abstract to discuss the responses of stratospheric wave

activities in southern hemisphere to ozone recovery. The main contents we modified are shown as follows:

    a)   The newly added Section 4:

**Response of Antarctic stratospheric wave activity to ozone recovery**

[revised manuscript text omitted]

**FIG. R3.** (a) Time series (solid lines) of aera-weighted total column ozone (TCO) over 60°S to 90°S derived MERRA-2 (red) and SBUV (blue) dataset. The dashed lines represent linear regression of TCO. (b, d) The TCO trends in September during 1980-2000 (b) and 2001-2017 (d) derived from MERRA-2 dataset. The outermost latitude in Fig. R3c, d is 40°S. (c, e) The zonal mean ozone trend on latitude-pressure profile in September during 1980-2000 (c) and 2001-2017 (e) derived from MERRA-2 dataset. The stippled regions in Figs. R3b-e represent trends significant at/above the 90% confidence level. Data in 2002 are removed when trends, regressions and significances

are calculated in Fig. R3.

[Figure]

**FIG. R4.** Difference of horizontal ozone forcing field averaged from 1000 hPa to 1 hPa between O3sen and O3ctrl. The outermost latitude in Fig. R4a is 40°S. Zonal mean difference of ozone forcing fields (b) on latitude-pressure profile in the southern hemisphere between O3sen and O3ctrl.

[Figure]

**FIG. R5.** Difference of (a) zonally averaged zonal wind, (b) zonally averaged temperature, (c) refractive index, (d) $a^2\overline{q}_\varphi$, (e) $-[\dfrac{(\overline{u}\cos\varphi)_\varphi}{\cos\varphi}]_\varphi$ (hereafter uyy term),

(f) $-\dfrac{a^2 f^2}{\rho_0}(\rho_0\dfrac{\overline{u}_z}{N^2})_z$ (hereafter uzz term) between O3sen and O3ctrl. The stippled

regions represent the difference significant at/above 90% confidence level.

[Figure]

**FIG. R6.** Differences of (a) stratospheric E-P flux (arrows, units in horizontal and vertical components are $0.02 \times 10^7$ and $0.05 \times 10^5$ kg·s$^{-2}$, respectively) and its divergence (shadings) with their (b) wave-1 component and (c) wave-2 component between the sensitive experiment (O3sen) and the control experiment (O3ctrl). The stippled regions represent the mean differences of E-P flux divergence significant at/above the 90% confidence level. The green contours from outside to inside (corresponding to p=0.1, 0.05) represent the mean differences of vertical E-P flux significant at the 90% and 95% confidence levels, respectively. (d) Frequency distributions (pillars, blue for O3ctrl and orange for O3sen) of vertical E-P flux (Fz, area-weighted from 200 hPa to 10 hPa over 70°S-50°S) and it 5-point low-pass filtered fitting curves (solid lines, blue for O3ctrl and red for O3sen) derived from 100 ensemble members.

**3.  Responses to comments about inflection of SST time series**

The reviewer suggests that it is necessary to add SST time series in the manuscript and find the inflection of SST to compare it with shift of stratospheric wave activity. We find that the inflections of SST time series around 2000 exist in some regions, which is shown in Fig. R7.

Following the suggestions, we have replaced the Fig. 10 (Fig. 4 in the original manuscript) with Fig. R7 and revised the first paragraph of Section 5 in the revised manuscript. The modified contents are shown as follows:

In addition, there are inflections of SST time series over some regions around 2000. In the southern Indian ocean, SST show insignificant trend during 1980-2000 and significant warming trend during 2000-2017 (Fig. R7c). The subtropical Pacific ocean east of Australia is linked with the Pacific-Southern America (PSA) wave train (e.g. Shen et al., 2020b), and the SST there shows significant warming trend during 1980-2000 and insignificant trend during 2000-2017. The SST in southeast Pacific show insignificant trend during 1980-2000 and significant cooling during 2000-2017 (Fig. R7e). Trends of SST in southern Atlantic ocean are opposite during these two piecewise periods, showing significant cooling trend during 1980-2000 and significant warming trend during 2000-2017. In a word, the spatial pattern of SST trend during 2000-2017 is obviously different from that during 1980-2000 (Figs. R7a, b), which may affect the tropospheric wave sources.

[Figure]

**FIG. R7**. Trends of SST in September over (a) 1980-2000 and (b) 2000-2017 derived from ERSST v5 dataset. The stippled regions represent trends significant at/above the 90% confidence level. (c-f) Time series (blue solid lines) of SST during 1980-2017 over different regions (titles). The dashed lines represent linear regressions of SST time series on piecewise periods (1980-2000 and 2000-2017). The "trend1" and "trend2" labeled in Figs. c-f represent the trend coefficients and the corresponding significances (bracketed) over 1980-2000 and 2000-2017, respectively.

*Specific:*

*L94, "...SST trend". Change to "...changes in SST".*

**Response:** Thanks for the suggestion. It has been revised.

*L118, "BDC" is defined in the abstract. Should it be defined in the text?*

**Response:** Thanks for the suggestion. BDC is defined as Brewer-Dobson circulation in the text

*L148, what is H?*

**Response:** Thanks for the careful check. H is scale height. It is defined after Eq. (4).

*L161, add an "a" before the first "zonal", and a "the" before the second "zonal".*

**Response:** Thanks for the careful check. It has been revised

*L180-188, provide some references.*

**Response:** Thanks for the suggestion. The reference has been added before Eq. (7) and Eq. (8). The added reference is

Shirley, D., Stanley, W., & Daniel, C.: Statistics for Research (Third Edition), (p. 627),

Hoboken, New Jersey: John Wiley & Sons Inc., 2004.

*L372, "...the SST trends". Change to "...the change in SST".*

**Response:** Thanks for the suggestion. It has been revised.

*Be consistent. Equ. or Equation ?*

**Response:** Thanks for the careful check. We have changed "Equation" or "Equ." to "Eq." in our revised manuscript.

*Figs. 2a and 2b. leave a space before and after "time series" in the title.*

**Response:** Thanks for the suggestion. The Fig. 2 has been replotted.

*References:*

*The references are not arranged exactly in the alphabetical order by the author's name.*

**Response:** Thanks for the careful check. The references have been rearranged.

---

## Author Response (AR2)

**Response to Editor's Comment**

**Weakening of Antarctic Stratospheric Planetary in Early Austral Spring Since the Early 2000s: A Response to Sea Surface Temperature Trends**
**(ACP-2021-395)**

Yihang Hu, Wenshou Tian, Jiankai Zhang, Tao Wang and Mian Xu

November 2021

**Response to Editor**

*Dear authors, Please include any figures -- in the response to reviewers but not yet included in the supplement -- in the supplement itself, so that ACP readers will have access to them. Then, modify main text to match numbering of the revised supplement. Examples: R1(d-f), R2, R3, R8, R9. The last two of these are especially important. Many thanks for your diligent efforts thus far. The manuscript will be acceptable for ACP after these minor revisions. –TD*

**Response:**

Thank you very much for further helpful comments. We have added all the figures and the table in the reply file (Fig. R1(d-f), Fig. R2, Fig. R3, Fig. R8, Fig. 9 and Table R2) to the supplementary file (corresponding to Fig. S10(a-c), Fig. S9, Fig. S10(d-i), Fig. S11, Fig. S4 and Table S2).

We also modify the relevant text in the manuscript to match them. The modifications which are highlight in blue are listed below:

1.   About L488-L490:

"The results of these experiments are summarized and displayed in Figure 14, which are quantified by the frequency distribution of southern hemisphere stratospheric vertical wave flux derived from the 100 ensemble members of each experiment."

change to:

"The results of stratospheric vertical wave flux over 50°S-70°S derived from the 100 ensemble members of each experiment are shown in Figure S4, and the frequency distributions of them are displayed in Figure 14."

2.   The last paragraph in section 7 is expanded into two paragraph:

"The southern hemisphere stratospheric wave activity trend from the early 1980s to the early 2000s has been investigated by Hu and Fu (2009) (hereafter HF2009) and hence is not discussed in detail in the above. HF2009 attributed the strengthening of stratospheric wave activity in austral spring during 1979-2006 to the SST trends as well, however, they gave no more details about the trends of tropospheric wave sources. In

this study, trends of tropospheric wave sources in September during 1980-2000 derived from MERRA-2 data is analyzed, and we also conducted an experiment (sstSHtrop80) forced by the changes of September SST during 1980-2000 over 20°N-70°S (see Fig. S9 for applied SST anomalies). The model result indicates that the SST changes over 20°N-70°S contribute to intensification of wave-2 component of tropospheric wave sources (Fig. S10f) and weakening of the wave-1 component (Fig. S10e), which is overall analogous to the trends derived from MERRA-2 data (Figs. S10b, c). Accordingly, the simulated wave-2 component of wave flux increases significantly in the stratosphere (Fig. S10h), while the response of the wave-1 component is not significant (Fig. S10i). In a word, the results from sstSHtrop80 suggest that the SST changes over 20°N-70°S induce a strengthening of stratospheric wave activity in September during 1980-2000. But it cannot explain the intensified wave-1 component of the stratospheric wave activity shown in Fig. 1b. A more detailed attribution of the trend of Antarctic stratospheric wave activity during 1980-2000 needs much more efforts.

The simulated stratospheric eddy heat flux (Fig. 11b in HF2009) forced by observed time-varying SST in HF2009 is relatively weak compared to that derived from reanalysis data (Fig. 6b in HF2009). Similarly, Wang and Waugh (2012) (hereafter WW2012) used stratosphere-resolving chemistry-climate model forced by time-varying factors to evaluate the trends of stratospheric temperature, residual circulation as well as wave activity during recent decades, and the trend of cumulative eddy heat flux shown in their paper is not significant (Fig. 6 in WW2012). Additionally, Polvani et al. (2018) used time-varying ODSs that cover the period from 1960s to 2080s to simulate Brewer-Dobson circulation and attained an obvious trend transition around 2000. We had also tried to conduct transient experiments forced by time-varying SST derived from ERSST v5 with different initial conditions. However, the trends of wave activities in the transient simulations are so weak, though opposite trend signs exist during 1980-2000 and 2000-2018 (Table S2, Fig. S11). The significance of simulated trend may be related to model performance and the length of simulating period. As the period we focus is relatively short and our purpose is attribution rather than generating

a real trend, we perform the ensemble time-slice experiments in this study, which are also used in many other previous researches (e.g., Hu et al., 2018; Kang et al., 2011; Zhang et al., 2016) to attribute trends in the atmosphere. In addition, most of the current climate models cannot generate a realistic wave activity trend as waves in the atmosphere are linked with various processes and factors (e.g., Baldwin & Dunkerton, 2005; Garcia & Randel, 2008; Labitzke, 2005; Shindell et al., 1999; Shu et al., 2013; Xie et al., 2008)."